# CREATING TRAINING SETS VIA WEAK INDIRECT SUPERVISION

**Jieyu Zhang**[1,2]**, Bohan Wang**[1,3]**, Xiangchen Song**[4]**, Yujing Wang**[1]**, Yaming Yang**[1]**, Jing Bai**[1]**, Alexander Ratner**[2,5]

[1]Microsoft Research Asia    [2]University of Washington
[3]University of Science and Technology of China    [4]Carnegie Mellon University    [5]Snorkel AI, Inc.
```
{jieyuz2, ajratner}@cs.washington.edu
{yujwang, yayaming, jbai}@microsoft.com
wbhfy@mail.ustc.edu.cn
xiangchensong@cmu.edu
```

## ABSTRACT

Creating labeled training sets has become one of the major roadblocks in machine learning. To address this, recent *Weak Supervision (WS)* frameworks synthesize training labels from multiple potentially noisy supervision sources. However, existing frameworks are restricted to supervision sources that share the same output space as the target task. To extend the scope of usable sources, we formulate *Weak Indirect Supervision (WIS)*, a new research problem for automatically synthesizing training labels based on *indirect supervision sources* that have different output label spaces. To overcome the challenge of mismatched output spaces, we develop a probabilistic modeling approach, PLRM, which uses user-provided label relations to model and leverage indirect supervision sources. Moreover, we provide a theoretically-principled test of the distinguishability of PLRM for unseen labels, along with a generalization bound. On both image and text classification tasks as well as an industrial advertising application, we demonstrate the advantages of PLRM by outperforming baselines by a margin of 2%-9%.

## 1  INTRODUCTION

One of the greatest bottlenecks of using modern machine learning models is the need for substantial amounts of manually-labeled training data. In real-world applications, such manual annotations are typically time-consuming, labor-intensive and static. To reduce the efforts of annotation, researchers have proposed Weak Supervision (WS) frameworks (Ratner et al., 2016; 2018; 2019; Fu et al., 2020) for synthesizing labels from multiple *weak supervision sources*, *e.g.*, heuristics, knowledge bases, or pre-trained classifiers. These frameworks have been widely applied on various machine learning tasks (Dunnmon et al., 2020; Fries et al., 2021; Safranchik et al., 2020; Lison et al., 2020; Zhou et al., 2020; Hooper et al., 2021; Zhan et al., 2019; Varma et al., 2019) and industrial data (Bach et al., 2019). Among them, *data programming* (Ratner et al., 2016), one representative example that generalizes many approaches in the literature, represents weak supervision sources as *labeling functions (LFs)* and synthesizes training labels using Probabilistic Graphical Model (PGM).

Given both the increasing popularity of WS and the general increase in open-source availability of machine learning models and tools, there is a rising tide of available supervision sources that WS frameworks and practitioners could potentially leverage, including pre-trained machine learning models or prediction APIs (Chen et al., 2020; d'Andrea & Mintz, 2019; Yao et al., 2017). However, existing WS frameworks only utilize weak supervision sources with the same label space as the target task. This incompatibility largely limits the scope of usable sources, necessitating manual effort from domain experts to provide supervision for *unseen* labels. For example, consider target task of classifying {"*dog*", "*wolf*", "*cat*", "*lion*"} and a set of three weak supervision sources (e.g. trained classifiers or expert heuristics) with disjoint output spaces {"*caninae*", "*felidae*"}, {"*domestic animals*", "*wild animals*"} and {"*husky*", "*bengal cat*"} respectively. We call these types of sources *indirect supervision sources*. For concreteness, we follow the general convention of *data programming* (Ratner et al., 2016) and refer to these sources as *indirect labeling functions (ILFs)*.

Despite their apparent utility, existing weak supervision methods could not directly leverage such ILFs, as their output spaces have no overlap with the target one.

In this paper, we formulate a novel research problem that aims to leverage such ILFs automatically, minimizing the manual efforts to develop and deploy new models. We refer to this as the *Weak Indirect Supervision (WIS)* setting, a new Weak Supervision paradigm which leverages ILFs, along with the relational structures between individual labels, to automatically create training labels.

The key difficulty of leveraging ILFs is due to the mismatched label spaces. To overcome this, we introduce pairwise relations between individual labels to the WIS setup, which are often available in structured sources (e.g. off-the-shelf Knowledge Bases (Miller, 1995; Sinha et al., 2015; Dong et al., 2020) or large scale label hierarchies (Murty et al., 2017; The Gene Ontology Consortium, 2018; Partalas et al., 2015) for various domains), or can be provided by subject matter experts in far less time than generating entirely new sets of weak supervision sources. For example, in the aforementioned example, we could rely on a biological species ontology to see that the unseen labels "*dog*" and "*cat*" are both subsumed by the seen label "*domestic animals*". Based on the label relations, we can automatically leverage the supervision sources as ILFs. Notably, previous work (Qu et al., 2020) also leveraged a label relation graph but was focused on relation extraction task in a few-shot learning setting, while You et al. (2020) proposed to learn label relations given data for each label in a transfer learning scenario. In contrast, we aim to solve the target task directly and without clean labeled data.

The remaining questions are (1) *how to synthesize labels based on pair-wise label relations and ILFs?* and (2) *How can we know whether, given a set of ILFs and label relations, the unseen labels are distinguishable or not?* To address the first question, we develop a *probabilistic label relation model (PLRM)*, the first PGM for WIS which aggregates the output of ILFs and models the label relations as dependencies between random variables. In turn, we use the learned PLRM to produce labels for training an end model. Furthermore, we derive the generalization error bound of PLRM based on assumptions similar to previous work (Ratner et al., 2016).

The second question presents an important stumbling block when dealing with unseen labels, as we may not be able to distinguish the unseen labels given existing label relations and ILFs, resulting in an unsatisfactory synthesized training set. To address this issue, we formally introduce the notion of *distinguishability* in WIS setting and theoretically establish an equivalence between: (1) the distinguishability of the label relation structure as well as the ILFs, and (2) the capability of PLRM to distinguish unseen labels. This result then leads to a simple sanity test for preventing the model from failing to distinguish unseen labels. In preliminary experiments, we observe a significant drop in model performance when the condition is violated.

In experiments, we make non-trivial adaptations for baselines from related settings to the new WIS problem. On both text and image classification tasks, we demonstrate the advantages of PLRM over adapted baselines. Finally, in a commercial advertising system where developers need to collect annotations for new ads tags, we illustrate how to formulate the training label collection as a WIS problem and apply PLRM to achieve an effective performance.

**Summary of Contributions.** Our contributions are summarized as follows:

- We formulate Weak Indirect Supervision (WIS), a new research problem which synthesizes training labels based on indirect supervision sources and label relations, minimizing human efforts of both data annotation and weak supervision sources construction;

- We develop the first model for WIS, the Probabilistic Label Relation Model (PLRM) with comparable statistical efficiency to previous WS frameworks and standard supervised learning;

- We introduce a new notion of distinguishability in WIS setting, and provide a simple test of the distinguishability of PLRM for unseen labels by theoretically establishing the connection between the label relation structures and distinguishability;

- We showcase the potential of the WIS formulation and the effectiveness of PLRM in a commercial advertising system for synthesizing training labels of new ads tags. On academic image and text classification tasks, we demonstrate the advantages of PLRM over baselines by quantitative experiments. Overall, PLRM outperforms baselines by a margin of 2%-9%.

## 2 RELATED WORK

Table 1: Comparisons between the proposed *weak indirect supervision (WIS)* and related machine learning tasks. Compared to normal and weakly supervised learning, WIS handles mismatched train and test label spaces. WIS is similar in spirit to indirect supervision (IS) and zero-shot learning (ZSL), but distinct in that WIS only takes as input weak or noisy labels and a simple set of logical label relations, and aims to output a training data set rather than a trained model, affording complete modularity in which final model class is used.

| Task | Label Type | $\mathcal{Y}_{train} = \mathcal{Y}_{test}$ | Label Information | When Label Info. is Required |
|---|---|:---:|:---:|:---:|
| **Supervised Learning (SL)** | Clean Labels | ✓ | – | – |
| **Weak Supervision (WS)** | Noisy Sources | ✓ | – | – |
| **Indirect Supervision (IS)** | Clean Labels | | Label Trans. Matrix | Training |
| **Zero-Shot Learning (ZSL)** | Clean Labels | | Label Embed. / Attribute | Training & Test |
| **Weak Indirect Supervision (WIS)** | Noisy Sources | | Label Relation | Training |

We briefly review related settings. The comparison between WIS and related tasks is in Table 1.

**Weak Supervision:** We draw motivation from recent work which model and integrate weak supervision sources using PGMs (Ratner et al., 2016; 2018; 2019; Fu et al., 2020) and other methods (Guan et al., 2018; Khetan et al., 2018) to create training sets. While they assume supervision sources share the same label space as the new tasks, we aim to leverage indirect supervision sources with mismatched label spaces in a labor-free way.

**Indirect Supervision:** Indirect supervision arises more generally in latent-variable models for various domains (Brown et al., 1993; Liang et al., 2013; Quattoni et al., 2004; Chang et al., 2010; Zhang et al., 2019). Very recently, Raghunathan et al. (2016) proposed to use the linear moment method for indirect supervision, wherein the *transition* between desired label space $\mathcal{Y}$ and indirect supervision space $\mathcal{O}$ is known, as well as the ground truth of indirect supervisions for training. In contrast, both are unavailable in WIS. Theoretically, Wang et al. (2020) developed a unified framework for analyzing the learnability of indirect supervision with shared or superset label spaces, while we focus on *disjoint* label spaces and the consequent unique challenge of *distinguishability* of unseen classes.

**Zero-Shot Learning:** Zero-Shot Learning (ZSL) (Lampert et al., 2009; Wang et al., 2019) aims to learn a classifier that is able to generalize to unseen classes. The WIS problem differentiates from ZSL by (1) in ZSL setting, the training and test data belong to seen and unseen classes, respectively, and training data is labeled, while for WIS, both training and test data belong to unseen classes and unlabeled; (2) ZSL tends to render a classifier that could predict unseen classes given certain label information, *e.g.*, label attributes (Romera-Paredes & Torr, 2015), label descriptions (Srivastava et al., 2018) or label similarities (Frome et al., 2013), while WIS aims to provide training labels for unlabeled training data, allowing users to train *any* machine learning models, and the label relations are used only in synthesizing training labels.

## 3 PRELIMINARY: WEAK SUPERVISION

We first describe the Weak Supervision (WS) setting. A glossary of notations used is in App. A.

**Definitions and notations.** We assume a $k$-way classification task, and have an *unlabeled* dataset $D$ consisting of $m$ data points. Denote by $X_i \in \mathcal{X}$ the individual data point and $Y_i \in \mathcal{Y} = \{y_1, \dots, y_k\}$ the *unobserved* interested label of $X_i$. We also have $n$ sources, each represented by a labeling function (LF) and denoted by $\lambda_j$. Each $\lambda_j$ outputs a label $\hat{Y}_i^j \in \mathcal{Y}_{\lambda_j} = \{\hat{y}_1^j, \dots, \hat{y}_{k_{\lambda_j}}^j\}$ on $X_i$, where $\mathcal{Y}_{\lambda_j}$ is the label space associated with $\lambda_j$ and $|\mathcal{Y}_{\lambda_j}| = k_{\lambda_j}$. We denote the concatenation of LFs' output as $\hat{Y}_i = [\hat{Y}_i^1, \hat{Y}_i^2, \dots, \hat{Y}_i^n]$, and the union set of LFs' label spaces as $\hat{\mathcal{Y}}$ with $|\hat{\mathcal{Y}}| = \hat{k}$. Note that $\hat{k}$ is not necessarily equal to the sum over $k_{\lambda_j}$, since LFs may have overlapping label spaces. We call $\hat{y} \in \hat{\mathcal{Y}}$ *seen* label and $y \in \mathcal{Y}$ *desired* labels. In WS settings, we have $\mathcal{Y} \subset \hat{\mathcal{Y}}$. Notably, we assume all the involved labels come from the same semantic space.

**The goal of WS.** The goal is to infer the training labels for the dataset $D$ based on LFs, and to use them to train an *end* discriminative classifier $f_W : \mathcal{X} \to \mathcal{Y}$, *all without ground truth training labels*.

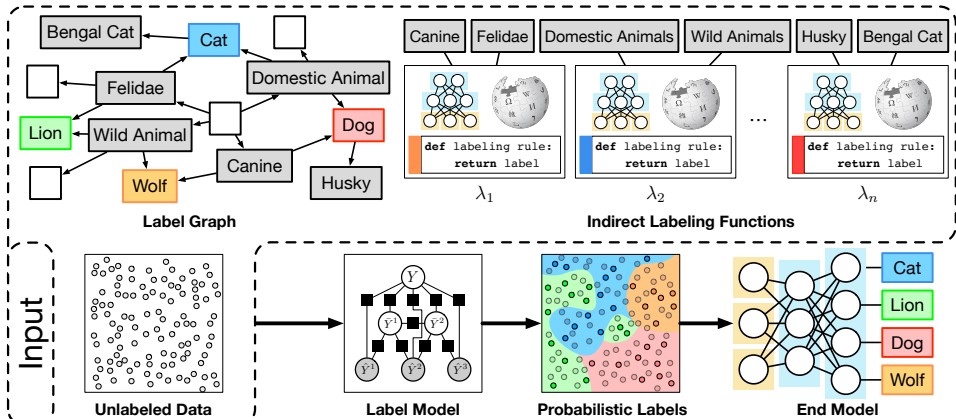

Figure 1: **An example of WIS problem:** the input consists of an unlabeled dataset, a label graph, and $n$ indirect labeling functions (ILFs). The ILFs represent weak supervision sources such as pretrained classifiers, knowledge bases, heuristic rules, etc. We can see that the ILFs cannot predict desired labels *i.e.*, {"*dog*", "*wolf*", "*cat*", "*lion*"}. To address this, a label graph is given; here we only visualize the subsuming relation. Finally, a label model, instantiated as a PGM, takes the ILF's outputs and produces probabilistic labels in the target output space, which are in turn used to train an end machine learning model that can generalize beyond them.

## 4 WEAK INDIRECT SUPERVISION

Now, we introduce the new Weak Indirect Supervision (WIS) problem. Unlike the standard WS setting, we only have indirect labeling functions (ILFs) instead of LFs, and an additional label graph is given. The goal of WIS remains the same as WS. An example of WIS problem is in Fig. 1.

**Indirect Labeling Function.** In WIS, we only have indirect labeling functions (ILFs), which cannot directly predict any desired labels, *i.e.*, $\hat{\mathcal{Y}} \cap \mathcal{Y} = \emptyset$. Therefore, we refer to the desired labels as *unseen* labels. To make it possible to leverage the ILFs, a label graph is given, which encodes pair-wise label relations between different seen and unseen labels.

**Label Graph.** Concretely, a label graph $G = (\mathcal{V}, \mathcal{E})$ consists of (1) a set of all the labels as nodes, *i.e.*, $\mathcal{V} = \hat{\mathcal{Y}} \cup \mathcal{Y}$, and (2) a set of pair-wise label relations as typed edges, *i.e.*, $\mathcal{E} = \{(y_i, y_j, t_{y_i y_j}) | t_{y_i y_j} \in \mathcal{T}, i < j, \forall y_i, y_j \in \mathcal{V}\}$. Here, $\mathcal{T}$ is the set of label relation types and, similar to Deng et al. (2014), there are four types of label relations: *exclusive*, *overlapping*, *subsuming*, *subsumed*, notated by $t^o, t^e, t^{sg}, t^{sd}$, respectively. Notably, for any *ordered* pair of labels $(y_i, y_j)$, their label relation should fall into *one* of the four types. The rationale behind these label relations is that when treating each label as a set, there are four unique set relations and each corresponds to one defined label relation respectively as shown in Fig. 2. For convenience, we denote the set of *non-exclusive neighbors* of a given label $y$ in $\hat{\mathcal{Y}}$ as $\mathcal{N}(y, \hat{\mathcal{Y}})$, *i.e.*, $\mathcal{N}(y, \hat{\mathcal{Y}}) = \{\hat{y} \in \hat{\mathcal{Y}} | t_{y\hat{y}} \neq t^e\}$.

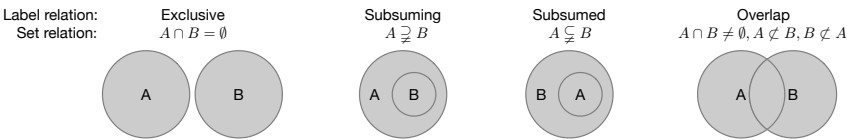

Figure 2: The one-to-one mapping between label relations and set relations.

## 5 PROBABILISTIC LABEL RELATION MODEL

One of the key difficulties in both WS and WIS is that we do not observe the true label $Y_i$. Following prior work (Ratner et al., 2016; 2019; Fu et al., 2020), we use a latent variable Probabilistic Graphical Model (PGM) for estimating $Y_i$ based on the $\hat{Y}_i$ output by ILFs. Specifically, the PGM is instantiated

as a factor graph model. This standard technique lets us describe the family of generative distributions in terms of $M$ known dependencies/factor functions $\{\phi\}$, and an unknown parameter $\Theta \in \mathbb{R}^M$ as $P_\Theta(\cdot) \propto \exp(\Theta^\top \Phi(\cdot))$, where $\Phi$ is the concatenation of $\{\phi\}$. However, the unique challenge for WIS is that the dependencies $\{\phi\}$ between $Y_i$ and $\hat{Y}_i$ are unknown due to the mismatches of label spaces. We overcome these by leveraging the label graph $G$ to build the dependencies for the PGM.

## 5.1 A Baseline PGM for WIS

In prior work (Ratner et al., 2016; Bach et al., 2017), the PGM for WS is governed by accuracy dependencies:
$$\phi_{y,j}^{\text{Acc}}(Y, \hat{Y}^j) := \mathbb{1}\{Y = \hat{Y}^j = y\}$$
which is defined for each $\lambda_j$ and $y \in \mathcal{Y}_{\lambda_j} \cap \mathcal{Y}$. However, in WIS, the ILFs cannot predict desired label $y \in \mathcal{Y}$. As a simple baseline approach to start, we leverage the coarse-grained exclusive/non-exclusive label relation to build a corresponding "accuracy" factor. Specifically, for an ILF $\lambda_j$ and one label $\hat{y} \in \mathcal{Y}_{\lambda_j}$, given a desired label $y \in \mathcal{Y}$, if $\hat{y}$ and $y$ have non-exclusive label relation, *i.e.*, $\hat{y} \in \mathcal{N}(y, \mathcal{Y}_{\lambda_j})$ we expect a certain portion of data assigned $\hat{y}$ should be labeled as $y$. Thus, we treat $\hat{Y}^j = \hat{y}$ as a pseudo indicator of $Y = y$ and add a pseudo accuracy dependency between them:
$$\phi_{y,\hat{y},j}^{\text{Acc}}(Y, \hat{Y}^j) := \mathbb{1}\{Y = y \wedge \hat{Y}^j = \hat{y}\}$$

We call the PGM governed by pseudo accuracy dependencies Weak Supervision with Label Graph (WS-LG). Notably, it can be treated as a simple adaptation of PGM for WS (Ratner et al., 2016; 2019; Fu et al., 2020) to the WIS problem. However, such a naïve adaptation might have two drawbacks:

1. It does not model specific dependencies ILFs with different undesired labels. For example, two ILFs outputting "*Husky*" and "*bulldog*" respectively would be naively modeled the same as if they both output "*Dog*".

2. It can only directly model exclusive/non-exclusive label relations, ignoring the prior knowledge encoded in other relation types, *i.e.*, subsuming, subsumed, or overlapping. For example, given an unseen label "*Dog*" and some ILFs outputting "*Husky*" or "*Domestic Animals*", WS-LG would treat all ILFs as indicators of "*Dog*". However, we know a "*Husky*" is of course a "*Dog*" (subsumed relation) while a "*Domestic Animals*" is not necessarily a "*Dog*" (subsuming relation).

## 5.2 Probabilistic Label Relation Model

To more directly model the full range and nuance of label relations, we propose a new *probabilistic label relation model (PLRM)*. In PLRM, we explicitly model both (1) the dependency between ILF outputs and the true labels in their output spaces, i.e. their direct accuracy, and (2) the dependencies between these labels and the target unseen labels, as separate dependency types, thus explicitly incorporating the full label relation graph into our model and learning its corresponding weights.

Concretely, we augment the WS-LG model with (1) latent variables representing the assignment of the data to each seen label, and (2) label relation dependencies which capture fine-grained label relations between these output labels and desired labels. To model seen label in $\hat{\mathcal{Y}}$, we introduce a binary latent random vector $\bar{Y} = [\bar{Y}^1, \ldots, \bar{Y}^{\hat{k}}]$, where $\bar{Y}^i$ indicating whether the data should be assigned $\hat{y}_i$. Then, for ILF $\lambda_j$ that could predict $\hat{y}_i$, we have accuracy dependency:

$$\phi_{\hat{y}_i,j}^{\text{Acc}}(\bar{Y}^i, \hat{Y}^j) := \mathbb{1}\{\bar{Y}^i = 1 \wedge \hat{Y}^j = \hat{y}_i\}$$

To model fine-grained label relations, for a desired label $y \in \mathcal{Y}$ and seen label $\hat{y}_i \in \hat{\mathcal{Y}}$, we add *label relation* dependencies. We enumerate the label relation dependencies corresponding to the four label relation types, *i.e.*, exclusive, overlapping, subsuming, subsumed, as follows:

$$\phi_{y,\hat{y}_i}^{e}(Y, \bar{Y}^i) := -\mathbb{1}\{Y = y \wedge \bar{Y}^i = 1\}$$

$$\phi_{y,\hat{y}_i}^{o}(Y, \bar{Y}^i) := \mathbb{1}\{Y = y \wedge \bar{Y}^i = 1\}$$

$$\phi_{y,\hat{y}_i}^{sg}(Y, \bar{Y}^i) := -\mathbb{1}\{Y \neq y \wedge \bar{Y}^i = 1\}$$

$$\phi_{y,\hat{y}_i}^{sd}(Y, \bar{Y}^i) := -\mathbb{1}\{Y = y \wedge \bar{Y}^i = 0\}$$

The above dependencies encode the prior knowledge of the label relations, but also allow the model to learn corresponding parameters. For example, an exclusive label relation dependency $\phi^e$ outputs -1 when two exclusive labels are activated at the same time for the same data, otherwise 0, which reflects our prior knowledge of the exclusive label relation; and the corresponding parameter can be treated as the *strength* of the label relation. Likewise, for any pair of seen labels, we add label relation dependency following the same convention. Finally, we specify the model as:

$$P_\Theta(Y, \bar{Y}, \hat{Y}) \propto \exp\left(\Theta^\top \Phi(Y, \bar{Y}, \hat{Y})\right) .$$ (1)

Recall that $Y$ is the unobserved true label, $\bar{Y}$ is the binary random vector, each of whose binary value $\bar{Y}^i$ reflects whether the data should be assigned seen label $\hat{y}_i \in \hat{\mathcal{Y}}$, and $\hat{Y}$ is the concatenated outputs of ILFs.

**Learning Objective.** We estimate the parameters $\hat{\Theta}$ by minimizing the negative log marginal likelihood $P_\Theta(\hat{Y})$ for observed ILF outputs $\hat{Y}_{1:m}$:

$$\hat{\Theta} = \arg\min_\Theta \ -\sum_{i=1}^m \log \sum_{Y,\bar{Y}} P_\Theta(Y, \bar{Y}, \hat{Y}_i) .$$ (2)

We follow Ratner et al. (2016) to optimize the objective using stochastic gradient descent.

**Training an End Model.** Let $p_{\hat{\Theta}}(Y \mid \hat{Y})$ be the probabilistic label (i.e. distribution) predicted by learned PLRM. We then train an end model $f_W : \mathcal{X} \to \mathcal{Y}$ parameterized by $W$, by minimizing the empirical *noise-aware loss* (Ratner et al., 2019) with respect to $\hat{\Theta}$ over $m$ unlabeled data points:

$$\hat{W} = \arg\min_W \frac{1}{m} \sum_{i=1}^m \mathbb{E}_{Y \sim p_{\hat{\Theta}}(Y|\hat{Y}_i)} \ell(Y, f_W(X_i)),$$ (3)

where $\ell(Y, f_W(X_i))$ is a standard cross entropy loss.

**Generalization Error Bound.** We extend previous results from (Ratner et al., 2016) to bound both the expected error of learned parameter $\hat{\Theta}$ and the expected risk for $\hat{W}$. All the proof details and description of assumptions can be found in Appendix.

**Theorem 1.** *Suppose that we run stochastic gradient descent to produce $\hat{\Theta}$ and $\hat{W}$ based on Eqs. (2) and (3), respectively, and that our setup satisfies certain assumptions (App D.2). Let $|D|$ be the size of the unlabeled dataset. Then we have*

$$\mathbb{E}\left\|\hat{\Theta} - \Theta^*\right\|^2 \leq O\left(M\frac{\log|D|}{|D|}\right), \quad \mathbb{E}\left[\ell(\hat{W}) - \ell(W^*)\right] \leq \chi + O\left(H\sqrt{\frac{\log|D|}{|D|}}\right) .$$

**Interpreting the Bound.** By Theorem 1, the two errors decrease by the rate $\tilde{O}(1/|D|)$ and $\tilde{O}(1/|D|^{1/2})$ respectively as $|D|$ increases. This shows that although we trade computational efficiency for the reduction of human efforts by using complex dependencies and more latent variables, we maintain comparable statistical efficiency as previous WS frameworks and supervised learning theoretically.

## 6 Distinguishability of Unseen Labels

One unique challenge of WIS is that there may exist pairs of unseen labels which cannot be distinguished by the learned model. For example, as shown in Fig. 3, where "*Dog*" is a seen label for which LFs could predict for and "*Husky*" and "*Bulldog*" are unseen labels for which we want to generate training labels; however, we could not distinguish between "*Husky*" and "*Bulldog*" even though the LFs make correct predictions of seen label "*Dog*", because both "*Husky*" and "*Bulldog*" share the same label relation to "*Dog*".

To tackle this issue, we theoretically connect the distinguishability of unseen labels to the label relation structures and provide a testable

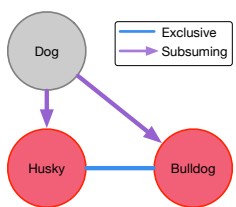

Figure 3: Example of indistinguishable unseen labels "*Husky*" and "*Bulldog*".

condition for the distinguishability. Intuitively, same label relation structures could lead to indistinguishable unseen labels as shown in Fig. 3; however, *it turns out to be challenging to prove that different label relation structures could guarantee the distinguishability with respect to the model*. To illustrate, we formally define the distinguishability as below.

**Definition 1** (Distinguishability). *For any model $P_\Theta(Y, \bar{Y}, \hat{Y})$ with parameters $\Theta$, any pair of unseen labels $y_i, y_j \in \mathcal{Y}$ are distinguishable w.r.t. the model, if for a.e. $\Theta > \mathbf{0}$ (element-wisely), there does NOT exist such a $\tilde{\Theta} > 0$ that, for $\forall \bar{Y}, \hat{Y}$, the following equations hold*

$$P_\Theta(Y = y_i | \bar{Y}, \hat{Y}) = P_{\tilde{\Theta}}(Y = y_j | \bar{Y}, \hat{Y}), P_\Theta(Y = y_j | \bar{Y}, \hat{Y}) = P_{\tilde{\Theta}}(Y = y_i | \bar{Y}, \hat{Y}), \quad (4)$$

$$P_\Theta(Y = y | \bar{Y}, \hat{Y}) = P_{\tilde{\Theta}}(Y = y | \bar{Y}, \hat{Y}), \forall y \in \mathcal{Y}/\{y_i, y_j\}, \quad (5)$$

$$P_\Theta(\hat{Y}) = P_{\tilde{\Theta}}(\hat{Y}). \quad (6)$$

From the definition, we can see that the opposite of distinguishability, *i.e.*, indistinguishability, describes an undesired model: for any learned parameter $\Theta > \mathbf{0}$, we can always find another $\tilde{\Theta}$ which optimizes the loss equally well (Eq. (6)), but Eqs. (4-5) implies *whenever $P_\Theta$ predict $y_i$, $P_{\tilde{\Theta}}$ will predict $y_j$ instead*, which reflects that the model cannot distinguish the two unseen labels. Note that the notion of distinguishability is different from the *identifiability* in PGMs: the *generic identifiability* (Allman et al., 2015), the strongest notion of identifiability, requires the model to be identifiable *up to label swapping*, while the distinguishability aims to avoid the label swapping.

However, distinguishability is hard to verify since Eqs. (4-5) and (6) need to hold for *any* possible configuration of $\bar{Y}, \hat{Y}$, and any pair of unseen labels. Fortunately, for the proposed PLRM, we prove that distinguishability is *equivalent* to the asymmetry of the label relation structures when two conditions hold. To state the required conditions, we first introduce the notations of consistency and informativeness to characterize the label graph and ILFs.

**Consistency.** We discuss the *consistency* of a label graph to avoid an ambiguous or unrealistic label graph. We interpret semantic labels $y_a, y_b$ as *sets $A$, $B$*, and then connect the label relations to the set relations (Fig. 2). Given the set interpretations, we define the *consistency* of label graph as:

**Definition 2** (Consistent Label Graph). *A label graph $G = (\mathcal{Y}, \mathcal{E})$ is consistent if the induced set relations are consistent.*

For example, assume $\mathcal{Y} = \{y_a, y_b, y_c\}$, and $t_{ab} = t_{bc} = t_{ca} = t^{sg}$. From $t_{ab}, t_{bc}$, we can observe that $A \supsetneq B \supsetneq C$, which contradicts to $C \not\supsetneq A$ implied by $t_{ca} = t^{sg}$. Thus, $G$ is inconsistent.

**Informativeness.** In addition, we try to describe what kind of ILF is desired. Intuitively, an ILF is uninformative if it always "votes" for one of the desired labels. For example, if the desired label space $\mathcal{Y}$ is {"*Dog*", "*Bird*"}, then for an ILF $\lambda_1$ outputting {"*Husky*", "*Bulldog*"}, we know "*Dog*" is non-exclusive to "*Husky*" and "*Bulldog*", while "*Bird*" exclusive to both. In such case, $\lambda_1$ can hardly provide information to help distinguish "*Dog*" from "*Bird*", because it always votes for "*Dog*". On the other hand, a binary classifier of "*Husky*", *i.e.*, $\lambda_2$, is favorable since it could output "*Not a Husky*" to avoid consistently voting for "*Dog*". We can see an undesired ILF always votes for a single desired label. To formally describe this, we define an *informative ILF* as:

**Definition 3** (Informative ILF). *An ILF $\lambda_j$ is informative if, for $\forall y \in \mathcal{Y}$, there exists $X_i \in \mathcal{D}$ s.t. the output of $\lambda_j$ on $X_i$ is not in $\mathcal{N}(y, \mathcal{Y}_{\lambda_j})$, i.e., $\hat{Y}_i^j \notin \mathcal{N}(y, \mathcal{Y}_{\lambda_j})$.*

**Testable Conditions for Distinguishability.** Based on the introduced notations, we prove the *necessary* and *sufficient* condition for learned PLRM being able to distinguish unseen labels:

**Theorem 2.** *For PLRM induced from a consistent label graph, as well as informative ILFs, for any pair of $y_i, y_j \in \mathcal{Y}$, they are indistinguishable, if and only if $t_{ik} = t_{jk}$ for $\forall y_k \in \hat{\mathcal{Y}}$.*

Theorem 2 provides users with a testable condition: *for any pair of unseen labels $y_i, y_j$, there should exist at least one seen label $y_k$ such that $y_k$ has different label relations to $y_i$ and $y_j$, i.e., $t_{ik} \neq t_{jk}$, so that PLRM is able to distinguish $y_i$ and $y_j$*. In preliminary experiments, we observe the violation of this condition causes a dramatic drop in overall performance (about 10 points). Notably, based on Theorem 2, users could theoretically guarantee the distinguishability of a pair of unseen labels by adding only one seen label and corresponding ILFs to break the symmetry.

# 7 EXPERIMENTS

We demonstrate the applicability and performance of our method on image classification tasks derived from ILSVRC2012 (Russakovsky et al., 2015) and text classification tasks derived from LSHTC-3 (Partalas et al., 2015). Both datasets have off-the-shelf label relation structure (Deng et al., 2014; Partalas et al., 2015), which are directed acyclic graphs (DAGS) and from which we could query pairwise label relations. Indeed, there is a one-to-one mapping between a DAG structure of labels and a consistent label graph (See App. E.1 for an example). The ILSVRC2012 dataset consists of 1.2M training images from 1,000 leave classes; for non-leave classes, we follow Deng et al. (2014) to aggregate images belonging to its descendent classes as its data points. The LSHTC-3 dataset consists of 456,886 documents and 36,504 labels organized in a DAG.

## 7.1 SETUP

For each dataset, we randomly sample 100 different label graphs, each of which consists of 8 classes, and use each label graph to construct a WIS task. For each label graph, we treat 3 of the sampled classes as unseen classes and the other 5 as seen classes. The distinguishable condition in Sec. 6 is ensured for all the WIS tasks, and the performance drop when it is violated can be found in App. G.1. We sample data belonging to unseen classes for our experiments and split them into train and test set. For image classification tasks, we follow Mazzetto et al. (2021b;a) to train a branch of image classifiers as supervision sources of seen classes. For text classification tasks, we made keyword-based labeling functions as supervision sources of seen classes following Zhang et al. (2021); each of the labeling functions returns its associated label when a certain keyword exists in the text, otherwise abstains. Notably, all the involved supervision sources are "weak" because they cannot predict the desired unseen classes. Experimental details and additional results are in App. F.

## 7.2 COMPARED METHODS AND RESULTS

In addition to the WS-LG baseline, which is an adaptation of Data Programming (Ratner et al., 2019) to WIS task, and PLRM, we also include the following baselines. Note that all compared methods input the same data, ILFs, and label relations throughout our experiments for fair comparisons.

**Label Relation Majority Voting (LR-MV).** We modify the majority voting method based on the label's non-exclusive neighbors: we replace $\hat{y}$ predicted by any ILF with the set of desired labels $\mathcal{N}(\hat{y}, \mathcal{Y})$, *i.e.*, the desired labels with non-exclusive relation to $\hat{y}$, then aggregate the modified votes.

**Weighted Label Relation Majority Voting (W-LR-MV).** LR-MV only leverages exclusive/non-exclusive label relations. To leverage fine-grained label relations, W-LR-MV attaches a *weight* to each replaced label. Specifically, if the ILF's output $\hat{y}$ is replaced with its *ancestor* label $y$ (subsumed relation), then the weight of $y$ equals 1, while for the other relations, the weight is $\frac{1}{|\mathcal{Y}^*(\hat{y})|}$, where $\mathcal{Y}^*(\hat{y}) = \{y \in \mathcal{Y}(\hat{y}) | t_{y\hat{y}} \neq t^{sd}\}$.

For the above methods, we compare the performance of (1) directly applying included models on the test set and (2) the end models (classifiers) trained with inferred training labels.

**Zero-Shot Learning (ZSL).** It is non-trivial to apply ZSL methods, because ZSL assumes label attributes for all classes and a labeled training set of seen classes, while WIS input an unlabeled dataset of unseen classes, label relations and ILFs. Fortunately, the Direct Attribute Prediction (DAP) (Lampert et al., 2013) method is able to make predictions solely based on attributes without labeled data, by training attribute classifier $p(a_i|x)$ for each attribute $a_i$. Therefore we include it in our experiments. The details of applying DAP can be found in App. F.2.

**Evaluation Results.** For a fair comparison, we fix the network architecture of the classifiers for all the methods. For image classification, we use ResNet-32 (He et al., 2016) and for text classification, we use logistic regression with pre-trained text embedding (Reimers & Gurevych, 2019). The overall results for both datasets can be found in Table 2. From the results, we can see that PLRM consistently outperforms baselines. The advantages of PLRM show the effect of not just leveraging the label graph, as the baselines do, but modeling the accuracy of ILFs and the strengths of label relations

Table 2: Averaged evaluation results over 100 WIS tasks derived from LSHTC-3 and ILSVRC2012.

| Method | | LSHTC-3 | | ILSVRC2012 | |
|---|---|---|---|---|---|
| | | Accuracy | F1-score | Accuracy | F1-score |
| DAP | | $42.90 \pm _{13.53}$ | $35.98 \pm _{15.73}$ | $33.25 \pm _{3.68}$ | $29.13 \pm _{4.63}$ |
| Label Model | LR-MV | $58.86 \pm _{10.50}$ | $54.33 \pm _{11.10}$ | $46.88 \pm _{10.66}$ | $40.11 \pm _{16.44}$ |
| | W-LR-MV | $59.28 \pm _{10.47}$ | $54.55 \pm _{11.36}$ | $41.39 \pm _{10.80}$ | $30.19 \pm _{16.94}$ |
| | WS-LG | $62.60 \pm _{10.12}$ | $57.50 \pm _{11.19}$ | $53.68 \pm _{7.62}$ | $52.15 \pm _{7.94}$ |
| | PLRM | $\mathbf{64.65} \pm _{11.30}$ | $\mathbf{60.01} \pm _{13.39}$ | $\mathbf{56.18} \pm _{7.35}$ | $\mathbf{54.94} \pm _{7.44}$ |
| End Model | LR-MV | $67.17 \pm _{12.25}$ | $62.49 \pm _{13.95}$ | $49.60 \pm _{12.80}$ | $42.83 \pm _{18.17}$ |
| | W-LR-MV | $66.57 \pm _{11.73}$ | $61.80 \pm _{13.24}$ | $42.61 \pm _{12.46}$ | $31.34 \pm _{18.20}$ |
| | WS-LG | $70.69 \pm _{13.05}$ | $67.36 \pm _{14.24}$ | $56.56 \pm _{9.68}$ | $54.57 \pm _{11.17}$ |
| | PLRM | $\mathbf{72.32} \pm _{13.18}$ | $\mathbf{69.37} \pm _{14.41}$ | $\mathbf{58.38} \pm _{8.27}$ | $\mathbf{56.83} \pm _{8.49}$ |

as PLRM does. The reported results have high variance, which actually indicates the 100 different WIS tasks are diverse and have varying difficulty. Also, we can see the end models are much better than directly applying the label models on the test set; this shows that the end models are able to generalize beyond the training labels produced by label models.

## 7.3 REAL-WORLD APPLICATION

In this section, on a commercial advertising system (CAS), we showcase how to reduce human annotation efforts of new labeling tasks by formulating them as WIS problems. In a CAS, ads tagging (classification) is a critical application for understanding the semantics of ads copy. When new ads and tags are added to the system, manual annotations need to be collected for training a new classifier. As tags are commonly organized as taxonomies, the label relations between existing and new tags are readily available or can be trivially figured out by humans; Existing classifiers and the heuristic rules previously used for annotating existing tags could serve as ILFs. Therefore, given (1) an unlabeled dataset of new tags, (2) the label relations, and (3) ILFs, we formulate it as a WIS problem.

On such WIS formulation, we apply our method and baselines, to synthesize training labels of new tags. Specifically, we have two WIS tasks where the tags are under the "*Car Accessories*" and "*Furniture*" categories respectively. For both tasks, we have 3 new tags and leverage 5 existing tags related to the new ones with given relations. On a test set, we evaluate the performance of DAP and the quality of labels produced by label models, as shown in Table 3. Note that since we re-use the existing labeling sources tailored for existing tags as ILFs and obtain label relations from an existing taxonomy, we achieve these results without any manual annotation or creation of new labeling functions. This demonstrates the potential of the proposed WIS task in real-world scenarios.

Table 3: Evaluation on product tagging with new tags.

| Category | Metric | DAP | LR-MV | W-LR-MV | WS-LG | PLRM |
|---|---|---|---|---|---|---|
| Car Accessories | F1 | 50.62 | 68.68 | 68.06 | 66.85 | **76.37** |
| | Accuracy | 52.83 | 68.17 | 67.67 | 66.33 | **75.83** |
| Furniture | F1 | 30.81 | 64.70 | 61.45 | 70.59 | **80.57** |
| | Accuracy | 33.60 | 72.53 | 72.13 | 74.51 | **82.02** |

## 8 CONCLUSION

We propose Weak Indirect Supervision (WIS), a new research problem which leverages indirect supervision sources and label relations to synthesize training labels for training machine learning models. We develop the first method for WIS called Probabilistic Label Relation Model (PLRM) with the generalization error bound of both PLRM and end model. We provide a theoretically-principled sanity test to ensure the distinguishability of unseen labels. Finally, we provide experiments to demonstrate the effectiveness of PLRM and its advantages over baselines on both academic datasets and industrial scenario.

**Reproducibility Statement.** All the assumptions and proofs of our theory can be found in App. C & D. Examples and illustrations of label graph are in App. E. Experimental details can be found in App. F. Additional experiments are in App. G.

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

SUPPLEMENTARY MATERIALS FOR
"CREATING TRAINING SETS VIA WEAK INDIRECT SUPERVISION"

The supplementary materials are organized as follows. In Appendix A, we provide a glossary of variables and symbols used in this paper. In Appendix B, we provide the details of PLRM model. In Appendix C and D, we provide the detailed proofs of Theorem 2 and Theorem 1 respectively. In Appendix E, we provide the detailed examples and illustrations of label graph in WIS. In Appendix F and G, we provide experimental details and additional experiment resulst respectively.

## A GLOSSARY OF SYMBOLS

Table 4: Glossary of variables and symbols used in this paper.

| Symbol | Simplified | Used for |
|---|---|---|
| $X_i$ | | The $i$-th data point, $X_i \in \mathcal{X}$ |
| $m$ | | Number of data points |
| $Y_i$ | | The true desired label of the $i$-th data point, $Y_i \in \mathcal{Y}$ |
| $y$ | | A semantic label, *e.g.*, "dog" |
| $\mathcal{Y}$ | | The set of desired labels, $\mathcal{Y} = \{y_1, y_2, \ldots, y_k\}$ |
| $k$ | | Cardinality of $\mathcal{Y}$, *i.e.*, $k = |\mathcal{Y}|$ |
| $\lambda_j$ | | The $j$-th Indirect labeling function (ILF) |
| $n$ | | Number of ILF |
| $\hat{Y}_i^j$ | | The output label of $j$-th ILF on $i$-th data point, $\hat{Y}_i^j \in \mathcal{Y}_{\lambda_j}$ |
| $\hat{Y}_i$ | | The concatenation of ILFs' output, $\hat{Y}_i = [\hat{Y}_i^1, \hat{Y}_i^2, \ldots, \hat{Y}_i^n]$ |
| $\hat{y}^j$ | | A semantic label in the label space of $\lambda_j$ |
| $\mathcal{Y}_{\lambda_j}$ | $\mathcal{Y}_j$ | Label label space of ILF $\lambda_j$, $\mathcal{Y}_{\lambda_j} = \{\hat{y}_1^j, \hat{y}_2^j, \ldots, \hat{y}_{k_{\lambda_j}}^j\}$ |
| $k_{\lambda_j}$ | $k_j$ | Cardinality of the output space of ILF $\lambda$, *i.e.*, $k_{\lambda_j} = |\mathcal{Y}_{\lambda_j}|$ |
| $\hat{\mathcal{Y}}$ | | Union set of all the $\mathcal{Y}_{\lambda_j}$, $\hat{\mathcal{Y}} = \{\hat{y}_1, \hat{y}_2, \ldots, \hat{y}_{\hat{k}}\}$ |
| $\hat{k}$ | | Cardinality of the $\hat{\mathcal{Y}}$, *i.e.*, $\hat{k} = |\hat{\mathcal{Y}}|$ |
| $K$ | | Total number of labels, *i.e.*, $K = \hat{k} + k$ |
| $\bar{Y}^i$ | | Latent binary variable indicating whether the data should be assigned $\hat{y}_i \in \hat{\mathcal{Y}}$. |
| $\bar{Y}$ | | Concatenation of all latent binary variable, $\bar{Y} = [\bar{Y}^1, \ldots, \bar{Y}^{\hat{k}}]$ |
| $G$ | | Label graph, $G = (\hat{\mathcal{Y}} \cup \mathcal{Y}, \mathcal{E})$ |
| $\mathcal{E}$ | | The set of label relations, $\mathcal{E} = \{(y_i, y_j, t_{y_i y_j}) | t_{y_i y_j} \in \mathcal{T}, i < j, \forall y_i, y_j \in \mathcal{V}\}$ |
| $\mathcal{T}$ | | The set of label relation types, $\mathcal{T} = \{t^e, t^o, t^{sd}, t^{sg}\}$ |
| $t^e$ | | Exclusive label relation |
| $t^o$ | | Overlap label relation |
| $t^{sg}$ | | Subsuming label relation |
| $t^{sd}$ | | Subsumed label relation |
| $\mathcal{N}(y, \hat{\mathcal{Y}})$ | | the set of non-exclusive neighbors of a given label $y$ in $\hat{\mathcal{Y}}$ |
| $\phi$ | | A single dependency, or, factor function |
| $\Phi$ | | Concatenation of all individual dependency |
| $M$ | | Number of total dependencies |
| $\theta$ | | A single parameter of the PGM |
| $\Theta$ | | Concatenation of all parameters of the PGM, $\Theta \in \mathbb{R}^M$ |
| $\hat{\Theta}$ | | The learned parameters |
| $\Theta^*$ | | The golden parameters |
| $W$ | | The parameter of an end model |
| $\hat{W}$ | | The learned parameters |
| $W^*$ | | The golden parameters |

# B  DETAILS OF THE PLRM

We use $Y, \bar{Y}$, and $\hat{Y}$ to represent random vector. Then, we give the formal form of the PLRM as:

$$P_\Theta(Y, \bar{Y}, \hat{Y}) \propto \exp\left(\Theta^\top \Phi(Y, \bar{Y}, \hat{Y})\right). \tag{7}$$

Recall that $Y$ is the unobserved true label, $\bar{Y}$ is the binary random vector, each of whose binary value $\bar{Y}^i$ reflects whether the data should be assigned seen label $\hat{y}_i \in \hat{\mathcal{Y}}$, and $\hat{Y}$ is the concatenated outputs of ILFs. Specifically, we enumerate $\Phi$ as below:

1.  (Pseudo accuracy dependency): $\forall j \in [n], y \in \mathcal{Y}/\{unknown\}, \hat{y} \in \mathcal{Y}_{\lambda_j}$, we have
    $$\phi^{Acc}_{y,\hat{y},j}(Y, \hat{Y}^j) := \mathbb{1}\{Y = y \wedge \hat{Y}^j = \hat{y} \wedge \hat{y} \in \mathcal{N}(y, \mathcal{Y}_{\lambda_j})\}^1$$

2.  (Accuracy dependency): $\forall j \in [n], \hat{y}_i \in \hat{\mathcal{Y}} \cap \mathcal{Y}_j$ we have
    $$\phi^{Acc}_{\hat{y}_i,j}(\bar{Y}^i, \hat{Y}^j) := \mathbb{1}\{\bar{Y}^i = 1 \wedge \hat{Y}^j = \hat{y}_i\}$$

3.  (Label relation dependency between seen labels): $\forall \hat{y}_i, \hat{y}_j \in \hat{\mathcal{Y}}, i < j$
    (a) if $t_{\hat{y}_i \hat{y}_j} = t^e$, we have
    $$\phi^e_{\hat{y}_i,\hat{y}_j}(\bar{Y}^i, \bar{Y}^j) := -\mathbb{1}\{\bar{Y}^i = 1 \wedge \bar{Y}^j = 1\}$$
    (b) if $t_{\hat{y}_i \hat{y}_j} = t^o$, we have
    $$\phi^o_{\hat{y}_i,\hat{y}_j}(\bar{Y}^i, \bar{Y}^j) := \mathbb{1}\{\bar{Y}^i = 1 \wedge \bar{Y}^j = 1\}$$
    (c) if $t_{\hat{y}_i \hat{y}_j} = t^{sg}$, we have
    $$\phi^{sg}_{\hat{y}_i,\hat{y}_j}(\bar{Y}^i, \bar{Y}^j) := -\mathbb{1}\{\bar{Y}^i = 0 \wedge \bar{Y}^j = 1\}$$
    (d) if $t_{\hat{y}_i \hat{y}_j} = t^{sd}$, we have
    $$\phi^{sd}_{\hat{y}_i,\hat{y}_j}(\bar{Y}^i, \bar{Y}^j) := -\mathbb{1}\{\bar{Y}^i = 1 \wedge \bar{Y}^j = 0\}$$

4.  (Label relation dependency between desired and seen labels): $\forall y \in \mathcal{Y}/\{unknown\}, \hat{y}_i \in \hat{\mathcal{Y}}$
    (a) if $t_{y\hat{y}_i} = t^e$, we have
    $$\phi^e_{y,\hat{y}_i}(Y, \bar{Y}^i) := -\mathbb{1}\{Y = y \wedge \bar{Y}^i = 1\}$$
    (b) if $t_{y\hat{y}_i} = t^o$, we have
    $$\phi^o_{y,\hat{y}_i}(Y, \bar{Y}^i) := \mathbb{1}\{Y = y \wedge \bar{Y}^i = 1\}$$
    (c) if $t_{y\hat{y}_i} = t^{sg}$, we have
    $$\phi^{sg}_{y,\hat{y}_i}(Y, \bar{Y}^i) := -\mathbb{1}\{Y \neq y \wedge \bar{Y}^i = 1\}$$
    (d) if $t_{y\hat{y}_i} = t^{sd}$, we have
    $$\phi^{sd}_{y,\hat{y}_i}(Y, \bar{Y}^i) := -\mathbb{1}\{Y = y \wedge \bar{Y}^i = 0\}$$

And example of our PLRM is shown in Fig. 4, where square with difference colors corresond to different dependency/factor functions in PLRM.

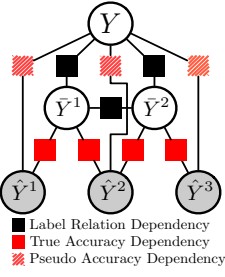

Figure 4: PLRM.

---

[1]When $\hat{y} \notin \mathcal{N}(y, \mathcal{Y}_{\lambda_j})$, $\phi^{Acc}_{y,\hat{y},j}$ is always zero and will not occur in the model. Here we use this form for the sack of rigorous representation.

## C  PROOF OF THEOREM 2

### C.1  SIMPLIFYING THE NOTATION

To simplify the indexing of dependencies, we use $\Phi^1$ to represent the concatenation of $\phi$ which involves both $Y$ and $\bar{Y}$, $\Phi^2$ to represent the concatenation of $\phi$ which involve both $\mathbf{Y}$ and $\hat{\mathbf{\Lambda}}$, and $\Phi^3$ to represent the concatenation of remaining $\phi$ which do not involve $\mathbf{Y}$.

Specifically, $\Phi^1$ consists of $k \times \hat{k}$ components corresponding to the dependency between the $k$ desired labels and the $\hat{k}$ seen labels. We use the subscript $i, j$ to denote the dependency function between the desired label $y_i$ and seen label $\hat{y}_j$, i.e.,

$$\Phi^1_{i,j} = \phi^?_{y_i, \hat{y}_j},$$

where ? is the corresponding relation.

Similarly, $\Phi^2$ consists of $k \times (\sum_{j=1}^n k_j)$ components corresponding to the dependency between the $k$ desired labels and the $k_j$ seen labels output by the ILF $\lambda_j$ ($j \in [n]$), and we use $\Phi^2_{i,j,l}$ to denote the dependency of $y_i$ and $\hat{y}^j_l$, and $\Phi^2_{i,j} = (\Phi^2_{i,j,l})_{l=1}^{k_j}$ to denote the dependency of $y_i$ and $\hat{y}^j$.

According to $\Phi^1$, $\Phi^2$, and $\Phi^3$, we also divide the parameter $\Theta$ into $\Theta^1$ (with elements being $\Theta^1_{i,j}$ correspondingly), $\Theta^2$ (with elements being $\Theta^2_{i,j} = (\Theta^2_{i,j,l})_{l=1}^{k_j}$ correspondingly), and $\Theta^3$, and the joint probability is then given as:

$$\mathbb{P}_\Theta(Y, \bar{Y}, \hat{Y}) = \frac{\exp\left((\Theta^1)^\mathsf{T} \Phi^1(Y, \bar{Y}) + (\Theta^2)^\mathsf{T} \Phi^2(Y, \hat{Y}) + (\Theta^3)^\mathsf{T} \Phi^3(\bar{Y}, \hat{Y})\right)}{\sum_{Y', \bar{Y}', \hat{Y}'} \exp\left((\Theta^1)^\mathsf{T} \Phi^1(Y', \bar{Y}') + (\Theta^2)^\mathsf{T} \Phi^2(Y', \hat{Y}') + (\Theta^3)^\mathsf{T} \Phi^3(\bar{Y}', \hat{Y}')\right)} \tag{8}$$

Also, for notation convenience, we adopt following simplifications:

1. $\forall y_i \in \mathcal{Y} \to \forall i \in [k]$ since $|\mathcal{Y}| = k$, similarly, $\forall \hat{y}_i \in \mathcal{Y}_j \to \forall i \in [k_j]$ and $\forall \hat{y}_i \in \hat{\mathcal{Y}} \to \forall i \in [\hat{k}]$;

2. $\forall \lambda_j \to \forall j \in [n]$ since we have $n$ ILFs in total;

3. $\phi^{t_{y_i y_j}}_{y_i, y_j} \to \phi^t_{y_i, y_j}$ where $t = t_{y_i y_j}$ and can be seen from the subscript of the dependency.

### C.2  PROPOSITIONS AND LEMMAS

First, we state some propositions and lemmas that will be useful in the proof to come.

**Proposition 1** (Multi-class classification). *For a multi-class classification task, $\forall y_i, y_j \in \mathcal{Y}$, we have $t_{y_i y_j} = t^e$. Similarly, $\forall \hat{y}_a, \hat{y}_b \in \hat{\mathcal{Y}}$, we have $t_{\hat{y}_a \hat{y}_b} = t^e$.*

**Lemma 1.** *For a consistent label graph $G$ and $\forall \hat{y}_l \in \hat{\mathcal{Y}}, \forall y_i, y_j \in \mathcal{Y}$, if $t_{y_i \hat{y}_l} = t^o$, we have $t_{y_j \hat{y}_l} \neq t^{sg}$ .*

*Proof.* For $\forall y_i, y_j \in \mathcal{Y}$, based on Proposition 1, we know $t_{y_i y_j} = t^e$, which implies (1) the intersection of the sets labeled by $y_i$ and $y_j$ is empty. For $\forall \hat{y}_l \in \hat{\mathcal{Y}}$, if $t_{y_i \hat{y}_l} = t^o$, we have (2) the intersection of the sets labeled by $y_i$ and $\hat{y}_l$ is not empty. If $t_{y_j \hat{y}_l} = t^{sg}$, which implies (3) $y_j \not\supseteq \hat{y}_l$. Based on (2)(3), we have $y_i \cap y_j \neq \emptyset$, which is contradictory to (1). Thus, we prove when $t_{y_i \hat{y}_l} = t^o, t_{y_j \hat{y}_l} \neq t^{sg}$. $\qquad\square$

**Lemma 2.** *For an informative ILF $\lambda_j$ and given any $y_d \in \mathcal{Y}$, there exists some $\hat{y}_l \in \mathcal{Y}_j$, such that, $\Phi^2_{d,j,l}(y_d, \hat{y}) = 0, \forall l \in [k_j]$.*

*Proof.* Because ILF $\lambda_j$ is informative, we know there exists one $\hat{y}_a \in \mathcal{Y}_j$ such that $\hat{y}_a$ is exclusive to $y_d$, i.e., $\hat{y}_a \notin \mathcal{N}(y_d, \mathcal{Y}_j)$. Therefore, for any $\hat{y}_l \in \hat{\mathcal{Y}}$, either $\hat{y}_a \neq \hat{y}_l$, or $\hat{y}_l = \hat{y}_a \notin \mathcal{Y}_j$, which leads to the conclusion by the definition of $\Phi^2_{d,j,l} = \phi^{\text{Acc}}_{y_d, \hat{y}_l, j}$. $\qquad\square$

## C.3 Definitions

Before the main proof, we connect the indistinguishablity of label relation structure with the dependency structure of PLRM by introducing the concept of *symmetry* as follows:

**Definition 4** (Symmetry). *For $y_i, y_j \in \mathcal{Y}$, we say $y_i$ and $y_j$ have symmetric dependency structure if the following equation holds:*

$$\Phi^1_{i,l} = \Phi^1_{j,l}, \forall l \in \hat{k};$$
$$\Phi^2_{i,a,b} = \Phi^2_{j,a,b}, \forall a \in [n], b \in [k_a]. \tag{9}$$

Based on the construction of PLRM, we know that for $\forall y_i, y_j \in \mathcal{Y}, \forall \hat{y}_b \in \hat{\mathcal{Y}}, t_{y_i \hat{y}_b} = t_{y_j \hat{y}_b}$ (the statement in Theorem 2) is *equivalent* to $y_i$ and $y_j$ have symmetric dependency structure.

## C.4 Equivalent Statement of Theorem 2

Our main result states that asymmetric is equivalent to distinguishable as in the following theorem, which can readily be seen to be identical to Theorem 2 in the main body of the paper:

**Theorem 3.** *For a probability model defined as Eq. (8) induced from a consistent label graph and informative ILFs, for any pair of $y_i, y_j \in \mathcal{Y}$, $y_i$ and $y_j$ are distinguishable if and only if they have asymmetric dependency structure.*

## C.5 Proof of the Necessity in Theorem 3: Necessary Condition

We first prove that for any $y_i, y_j \in \mathcal{Y}$, $y_i$ and $y_j$ have asymmetric dependency structure is the *necessary* condition of that they are distinguishable.

*Proof of Theorem 3.* We prove this theorem by reduction to absurdity. Suppose $y_i$ and $y_j$ are symmetric. Then, by Eq. (8), the distribution of $Y$ condition on any $\bar{Y}$ and $\hat{Y}$ can be calculated as follows:

$$\mathbb{P}_\Theta(Y = y_i | \bar{Y}, \hat{Y}) = \frac{\mathbb{P}_\Theta(y_i, \bar{Y}, \hat{Y})}{\mathbb{P}_\Theta(\bar{Y}, \hat{Y})}.$$

On the other hand, applying $Y = y_i$ in the definition of $\Phi^2$ leads to

$$\Phi^2_{r,a,l}(y_i, \cdot) = 0, \forall r \in [k], r \neq i, \forall a \in [n], \forall l \in [k_a].$$

We further separate $\Phi^1$ into $(\Phi^1_i)^k_{i=1}$, where $\Phi^1_i$ collects all the dependency in $\Phi^1$ with $y_i$ involved, i.e.,

$$\Phi^1_i = (\Phi^1_{i,j})^{\hat{k}}_{j=1},$$

with the corresponding parameters respectively denoted as $\Theta^1_i$ with $\Theta^1 = (\Theta^1_i)^k_{i=1}$. Similarly, $\Phi^2$ is also divided into $(\Phi^2_i)^k_{i=1}$ following the same routine and $\Theta^2$ is respectively divided into $(\Theta^2_i)^k_{i=1}$. Specifically, if $y_i$ and $y_j$ are symmetric, we further have

$$\Phi^1_i = \Phi^1_j, \Phi^2_i = \Phi^2_j.$$

Based on the notation, $\mathbb{P}_\Theta(Y = y_i | \bar{Y}, \hat{Y})$ can then be represented as

$$\mathbb{P}_\Theta(Y = y_i | \bar{Y}, \hat{Y}) \left( \sum_{Y'} \exp \left( (\Theta^1)^\mathsf{T} \Phi^1(Y', \bar{Y}) + (\Theta^2)^\mathsf{T} \Phi^2(Y', \hat{Y}) + (\Theta^3)^\mathsf{T} \Phi^3(\bar{Y}, \hat{Y}) \right) \right)$$
$$= \exp \left( \sum_{l=1}^k (\Theta^1_l)^\mathsf{T} \Phi^1_l(y_i, \bar{Y}) + \sum_{l=1}^k (\Theta^2_l)^\mathsf{T} \Phi^2_l(y_i, \hat{Y}) + (\Theta^3)^\mathsf{T} \Phi^3(\bar{Y}, \hat{Y}) \right)$$

which further leads to

$$\mathbb{P}_{\Theta}(Y = y_i | \bar{Y}, \hat{Y}) \left( \sum_{Y'} \exp\left( (\Theta^1)^\top \Phi^1(Y', \bar{Y}) + (\Theta^2)^\top \Phi^2(Y', \hat{Y}) \right) \right)$$
$$= \exp\left( \sum_{l=1}^{k} (\Theta_l^1)^\mathsf{T} \Phi_l^1(y_i, \bar{Y}) + (\Theta_i^2)^\mathsf{T} \Phi_i^2(y_i, \hat{Y}) \right) \tag{10}$$

which is independent of $\Theta^3$. Similarly,

$$\mathbb{P}_{\Theta}(Y = y_j | \bar{Y}, \hat{Y}) \left( \sum_{Y'} \exp\left( (\Theta^1)^\top \Phi^1(Y', \bar{Y}) + (\Theta^2)^\top \Phi^2(Y', \hat{Y}) \right) \right)$$
$$= \exp\left( \sum_{l=1}^{k} (\Theta_l^1)^\mathsf{T} \Phi_l^1(y_i, \bar{Y}) + (\Theta_j^2)^\mathsf{T} \Phi_j^2(y_j, \hat{Y}) \right) \tag{11}$$

and $\forall l \in [k]/\{i, j\}$,

$$\mathbb{P}_{\Theta}(Y = y_l | \bar{Y}, \hat{Y}) \left( \sum_{Y'} \exp\left( (\Theta^1)^\top \Phi^1(Y', \bar{Y}) + (\Theta^2)^\top \Phi^2(Y', \hat{Y}) \right) \right)$$
$$= \exp\left( \sum_{l=1}^{k} (\Theta_l^1)^\mathsf{T} \Phi_l^1(y_i, \bar{Y}) + (\Theta_l^2)^\mathsf{T} \Phi_l^2(y_l, \hat{Y}) \right) \tag{12}$$

Let $\tilde{\Theta}$ be defined as follows:

$$\tilde{\Theta}_i^1 = \Theta_j^1, \tilde{\Theta}_j^1 = \Theta_i^1, \tilde{\Theta}_l^1 = \Theta_l^1, \forall l \notin \{i, j\},$$
$$\tilde{\Theta}_i^2 = \Theta_j^2, \tilde{\Theta}_j^2 = \Theta_i^2, \tilde{\Theta}_l^2 = \Theta_l^2, \forall l \notin \{i, j\},$$

and

$$\tilde{\Theta}^3 = \Theta^3.$$

We then have

$$\frac{\mathbb{P}_{\Theta}(Y = y_i | \bar{Y}, \hat{Y})}{\mathbb{P}_{\tilde{\Theta}}(Y = y_j | \bar{Y}, \hat{Y})}$$
$$= \frac{\left( \sum_{Y'} \exp\left( (\tilde{\Theta}^1)^\top \Phi^1(Y', \bar{Y}) + (\tilde{\Theta}^2)^\top \Phi^2(Y', \hat{Y}) \right) \right)}{\left( \sum_{Y'} \exp\left( (\Theta^1)^\top \Phi^1(Y', \bar{Y}) + (\Theta^2)^\top \Phi^2(Y', \hat{Y}) \right) \right)}$$
$$\cdot \exp\left( (\Theta_i^1)^\mathsf{T} (\Phi_i^1(y_i, \bar{Y}) - \Phi_j^1(y_j, \bar{Y})) + (\Theta_j^1)^\mathsf{T} (\Phi_j^1(y_i, \bar{Y}) - \Phi_i^1(y_j, \bar{Y})) + (\Theta_i^2)^\mathsf{T} (\Phi_i^2(y_i, \hat{Y}) - \Phi_j^2(y_j, \hat{Y})) \right)$$
$$= \frac{\left( \sum_{Y'} \exp\left( (\tilde{\Theta}^1)^\top \Phi^1(Y', \bar{Y}) + (\tilde{\Theta}^2)^\top \Phi^2(Y', \hat{Y}) \right) \right)}{\left( \sum_{Y'} \exp\left( (\Theta^1)^\top \Phi^1(Y', \bar{Y}) + (\Theta^2)^\top \Phi^2(Y', \hat{Y}) \right) \right)}.$$

Similarly,

$$\frac{\mathbb{P}_{\Theta}(Y = y_j | \bar{Y}, \hat{Y})}{\mathbb{P}_{\tilde{\Theta}}(Y = y_i | \bar{Y}, \hat{Y})}$$
$$= \frac{\left( \sum_{Y'} \exp\left( (\tilde{\Theta}^1)^\top \Phi^1(Y', \bar{Y}) + (\tilde{\Theta}^2)^\top \Phi^2(Y', \hat{Y}) \right) \right)}{\left( \sum_{Y'} \exp\left( (\Theta^1)^\top \Phi^1(Y', \bar{Y}) + (\Theta^2)^\top \Phi^2(Y', \hat{Y}) \right) \right)}$$
$$\cdot \exp\left( (\Theta_j^1)^\mathsf{T} (\Phi_j^1(y_j, \bar{Y}) - \Phi_i^1(y_i, \bar{Y})) + (\Theta_i^1)^\mathsf{T} (\Phi_i^1(y_j, \bar{Y}) - \Phi_j^1(y_i, \bar{Y})) + (\Theta_j^2)^\mathsf{T} (\Phi_j^2(y_j, \hat{Y}) - \Phi_i^2(y_i, \hat{Y})) \right)$$
$$= \frac{\left( \sum_{Y'} \exp\left( (\tilde{\Theta}^1)^\top \Phi^1(Y', \bar{Y}) + (\tilde{\Theta}^2)^\top \Phi^2(Y', \hat{Y}) \right) \right)}{\left( \sum_{Y'} \exp\left( (\Theta^1)^\top \Phi^1(Y', \bar{Y}) + (\Theta^2)^\top \Phi^2(Y', \hat{Y}) \right) \right)}.$$

and $\forall l \in [k]/\{i,j\}$,

$$\frac{\mathbb{P}_\Theta(Y = y_l|\bar{Y}, \hat{Y})}{\mathbb{P}_{\tilde{\Theta}}(Y = y_l|\bar{Y}, \hat{Y})} = \frac{\left(\sum_{Y'} \exp\left((\tilde{\Theta}^1)^\top \Phi^1(Y', \bar{Y}) + (\tilde{\Theta}^2)^\top \Phi^2(Y', \hat{Y})\right)\right)}{\left(\sum_{Y'} \exp\left((\Theta^1)^\top \Phi^1(Y', \bar{Y}) + (\Theta^2)^\top \Phi^2(Y', \hat{Y})\right)\right)}.$$

Similarly, we have

$$\frac{\mathbb{P}_\Theta(Y = unknown|\bar{Y}, \hat{Y})}{\mathbb{P}_{\tilde{\Theta}}(Y = unknown|\bar{Y}, \hat{Y})} = \frac{\left(\sum_{Y'} \exp\left((\tilde{\Theta}^1)^\top \Phi^1(Y', \bar{Y}) + (\tilde{\Theta}^2)^\top \Phi^2(Y', \hat{Y})\right)\right)}{\left(\sum_{Y'} \exp\left((\Theta^1)^\top \Phi^1(Y', \bar{Y}) + (\Theta^2)^\top \Phi^2(Y', \hat{Y})\right)\right)}.$$

Therefore, we have

$$\frac{\mathbb{P}_\Theta(Y = y_i|\bar{Y}, \hat{Y})}{\mathbb{P}_{\tilde{\Theta}}(Y = y_j|\bar{Y}, \hat{Y})} = \frac{\mathbb{P}_\Theta(Y = y_j|\bar{Y}, \hat{Y})}{\mathbb{P}_{\tilde{\Theta}}(Y = y_i|\bar{Y}, \hat{Y})} = \frac{\mathbb{P}_\Theta(Y = y|\bar{Y}, \hat{Y})}{\mathbb{P}_{\tilde{\Theta}}(Y = y|\bar{Y}, \hat{Y})}, \forall y \in \mathcal{Y}/\{y_i, y_j\}.$$

Since

$$\mathbb{P}_\Theta(Y = y_i|\bar{Y}, \hat{Y}) + \mathbb{P}_\Theta(Y = y_j|\bar{Y}, \hat{Y}) + \sum_{l \neq i,j} \mathbb{P}_\Theta(Y = y_l|\bar{Y}, \hat{Y}) = 1,$$

and

$$\mathbb{P}_{\tilde{\Theta}}(Y = y_j|\bar{Y}, \hat{Y}) + \mathbb{P}_{\tilde{\Theta}}(Y = y_i|\bar{Y}, \hat{Y}) + \sum_{l \neq i,j} \mathbb{P}_{\tilde{\Theta}}(Y = y_l|\bar{Y}, \hat{Y}) = 1,$$

we obtain that

$$\mathbb{P}_\Theta(Y = y_i|\bar{Y}, \hat{Y}) = \mathbb{P}_{\tilde{\Theta}}(Y = y_j|\bar{Y}, \hat{Y})$$
$$\mathbb{P}_\Theta(Y = y_j|\bar{Y}, \hat{Y}) = \mathbb{P}_{\tilde{\Theta}}(Y = y_i|\bar{Y}, \hat{Y})$$
$$\mathbb{P}_\Theta(Y = y_l|\bar{Y}, \hat{Y}) = \mathbb{P}_{\tilde{\Theta}}(Y = y_l|\bar{Y}, \hat{Y}),$$

which indicates $y_i$ and $y_j$ indistinguishable, and leads to a contradictory.

The proof is completed. $\qquad\square$

## C.6 PROOF OF THEOREM 3: SUFFICIENT CONDITION

We then prove that for any $y_i, y_j \in \mathcal{Y}$, $y_i$ and $y_j$ have asymmetric dependency structure is the *sufficient* condition of that they are distinguishable.

*Proof.* We use the same notations $(\Theta_i^1)_{i=1}^k$, $(\Theta_i^2)_{i=1}^k$, and $\Theta^3$ in Appendix C.5 to denote the separation of the parameter $\Theta$. Let $\Theta$ be any parameter satisfying that there exists a parameter $\tilde{\Theta}$, such that Eq. (4-5) holds. By Eqs. (10), (11), and Eq. (12) together with Eqs. (4-5), we have $\forall r \in [k], r \neq i, j$,

$$\frac{\exp\left((\Theta_i^1)^\mathsf{T} \Phi_i^1(y_i, \bar{Y}) + (\Theta_j^1)^\mathsf{T} \Phi_j^1(y_i, \bar{Y}) + (\Theta_i^2)^\mathsf{T} \Phi_i^2(y_i, \hat{Y})\right)}{\exp\left((\tilde{\Theta}_i^1)^\mathsf{T} \Phi_i^1(y_j, \bar{Y}) + (\tilde{\Theta}_j^1)^\mathsf{T} \Phi_j^1(y_j, \bar{Y}) + (\tilde{\Theta}_j^2)^\mathsf{T} \Phi_j^2(y_j, \hat{Y})\right)}$$
$$=\frac{\exp\left((\Theta_i^1)^\mathsf{T} \Phi_i^1(y_j, \bar{Y}) + (\Theta_j^1)^\mathsf{T} \Phi_j^1(y_j, \bar{Y}) + (\Theta_j^2)^\mathsf{T} \Phi_j^2(y_j, \hat{Y})\right)}{\exp\left((\tilde{\Theta}_i^1)^\mathsf{T} \Phi_i^1(y_i, \bar{Y}) + (\tilde{\Theta}_j^1)^\mathsf{T} \Phi_j^1(y_i, \bar{Y}) + (\tilde{\Theta}_i^2)^\mathsf{T} \Phi_i^2(y_i, \hat{Y})\right)}$$
$$=\frac{\exp\left((\Theta_i^1)^\mathsf{T} \Phi_i^1(y_r, \bar{Y}) + (\Theta_j^1)^\mathsf{T} \Phi_j^1(y_r, \bar{Y})\right)}{\exp\left((\tilde{\Theta}_i^1)^\mathsf{T} \Phi_i^1(y_r, \bar{Y}) + (\tilde{\Theta}_j^1)^\mathsf{T} \Phi_j^1(y_r, \bar{Y})\right)} = \frac{\exp\left((\Theta_i^1)^\mathsf{T} \Phi_i^1(y_j, \bar{Y}) + (\Theta_j^1)^\mathsf{T} \Phi_j^1(y_i, \bar{Y})\right)}{\exp\left((\tilde{\Theta}_i^1)^\mathsf{T} \Phi_i^1(y_j, \bar{Y}) + (\tilde{\Theta}_j^1)^\mathsf{T} \Phi_j^1(y_i, \bar{Y})\right)}.$$

By simple rearranging, we have

$$((\Theta_i^1)^\mathsf{T} \Phi_i^1(y_i, \bar{Y}) + (\Theta_j^1)^\mathsf{T} \Phi_j^1(y_i, \bar{Y}) + (\Theta_i^2)^\mathsf{T} \Phi_i^2(y_i, \hat{Y}) + (\Theta_j^2)^\mathsf{T} \Phi_j^2(y_i, \hat{Y}))$$
$$- ((\tilde{\Theta}_i^1)^\mathsf{T} \Phi_i^1(y_j, \bar{Y}) + (\tilde{\Theta}_j^1)^\mathsf{T} \Phi_j^1(y_j, \bar{Y}) + (\tilde{\Theta}_i^2)^\mathsf{T} \Phi_i^2(y_j, \hat{Y}) + (\tilde{\Theta}_j^2)^\mathsf{T} \Phi_j^2(y_j, \hat{Y}))$$
$$=((\Theta_i^1)^\mathsf{T} \Phi_i^1(y_j, \bar{Y}) + (\Theta_j^1)^\mathsf{T} \Phi_j^1(y_j, \bar{Y}) + (\Theta_i^2)^\mathsf{T} \Phi_i^2(y_j, \hat{Y}) + (\Theta_j^2)^\mathsf{T} \Phi_j^2(y_j, \hat{Y}))$$
$$- ((\tilde{\Theta}_i^1)^\mathsf{T} \Phi_i^1(y_i, \bar{Y}) + (\tilde{\Theta}_j^1)^\mathsf{T} \Phi_j^1(y_i, \bar{Y}) + (\tilde{\Theta}_i^2)^\mathsf{T} \Phi_i^2(y_i, \hat{Y}) + (\tilde{\Theta}_j^2)^\mathsf{T} \Phi_j^2(y_i, \hat{Y}))$$
$$=((\Theta_i^1)^\mathsf{T} \Phi_i^1(y_j, \bar{Y}) + (\Theta_j^1)^\mathsf{T} \Phi_j^1(y_i, \bar{Y})) - ((\tilde{\Theta}_i^1)^\mathsf{T} \Phi_i^1(y_j, \bar{Y}) + (\tilde{\Theta}_j^1)^\mathsf{T} \Phi_j^1(y_i, \bar{Y})). \quad (13)$$

By the equality between the second term and the third term in Eq. (13), we obtain that

$$(\Theta_j^1)^\mathsf{T}\Phi_j^1(y_i, \bar{Y}) - (\tilde{\Theta}_i^1)^\mathsf{T}\Phi_i^1(y_j, \bar{Y})$$
$$= ((\Theta_j^1)^\mathsf{T}\Phi_j^1(y_j, \bar{Y}) + (\Theta_j^2)^\mathsf{T}\Phi_j^2(y_j, \hat{Y})) - ((\tilde{\Theta}_i^1)^\mathsf{T}\Phi_i^1(y_i, \bar{Y}) + (\tilde{\Theta}_i^2)^\mathsf{T}\Phi_i^2(y_i, \hat{Y})). \quad (14)$$

We further set $\bar{Y}$ in Eq. (14) respectively to $e_l$ (the one hot vector with its $l$-th position being 1) and $\mathbf{0}$ for any fixed $l \in [\hat{k}]$, i.e.,

$$((\Theta_j^1)^\mathsf{T}\Phi_j^1(y_i, e_l) - (\Theta_j^1)^\mathsf{T}\Phi_j^1(y_i, \mathbf{0})) - ((\tilde{\Theta}_i^1)^\mathsf{T}\Phi_i^1(y_j, e_l) - (\tilde{\Theta}_i^1)^\mathsf{T}\Phi_i^1(y_j, \mathbf{0}))$$
$$= ((\Theta_j^1)^\mathsf{T}\Phi_j^1(y_j, e_l) - (\Theta_j^1)^\mathsf{T}\Phi_j^1(y_j, \mathbf{0})) - ((\tilde{\Theta}_i^1)^\mathsf{T}\Phi_i^1(y_i, e_l) - (\tilde{\Theta}_i^1)^\mathsf{T}\Phi_i^1(y_i, \mathbf{0})),$$

which by simple rearranging further leads to

$$\Theta_{j,l}^1(\Phi_{j,l}^1(y_j, 1) - \Phi_{j,l}^1(y_j, 0) - \Phi_{j,l}^1(y_i, 1)) = \tilde{\Theta}_{i,l}^1(\Phi_{i,l}^1(y_i, 1) - \Phi_{i,l}^1(y_i, 0) - \Phi_{i,l}^1(y_j, 1)).$$

Since $\Theta_{j,l}^1, \tilde{\Theta}_{i,l}^1 > 0$, and by definition we have

$$|\Phi_{j,l}^1(y_j, 1) - \Phi_{j,l}^1(y_j, 0) - \Phi_{j,l}^1(y_i, 1)| = 1,$$

and

$$|\Phi_{i,l}^1(y_i, 1) - \Phi_{i,l}^1(y_i, 0) - \Phi_{i,l}^1(y_j, 1)| = 1,$$

we obtain $\Theta_{j,l}^1 = \tilde{\Theta}_{i,l}^1$, and

$$\Phi_{j,l}^1(y_j, 1) - \Phi_{j,l}^1(y_j, 0) - \Phi_{j,l}^1(y_i, 1) = \Phi_{i,l}^1(y_i, 1) - \Phi_{i,l}^1(y_i, 0) - \Phi_{i,l}^1(y_j, 1). \quad (15)$$

Therefore, either $t_{y_j\hat{y}_l} \in \{t^o, t^{sd}, t^{sg}\}$ and $t_{y_i\hat{y}_l} \in \{t^o, t^{sd}, t^{sg}\}$, or $t_{y_j\hat{y}_l} = t^e$ and $t_{y_i\hat{y}_l} = t^e$, which by definition further indicates that $\Phi_i^2 = \Phi_j^2$ (recall the way we build dependency between $Y$ and $\hat{Y}$).

As $l$ is arbitrarily picked, we then have $\Theta_j^1$ is equal to $\tilde{\Theta}_i^1$ component-wisely.

By the equality between the first term and the third term in Eq. (13) and following exact the same routine, we also have $\tilde{\Theta}_j^1 = \Theta_i^1$.

On the other hand, for any $r \in [\hat{k}]$, fixing $\bar{Y}$ and $\hat{Y}^s$ ($\forall s \neq r$), and setting $\hat{Y}_r = \hat{y}_l^r$ ($l \in k_r$, $\hat{y}_l^r \in \mathcal{N}(y_j, \mathcal{Y}_l)$) in Eq. (14), we have

$$(\Theta_j^1)^\mathsf{T}\Phi_j^1(y_j, \bar{Y}) + (\tilde{\Theta}_i^1)^\mathsf{T}\Phi_i^1(y_j, \bar{Y}) + \Theta_{j,r,l}^2\Phi_{j,r,l}^2(y_j, \hat{y}_l^r) + \sum_{s \neq r}\Theta_{j,s}^2\Phi_{j,s}^2(y_j, Y^s)$$
$$= (\Theta_j^1)^\mathsf{T}\Phi_j^1(y_i, \bar{Y}) + (\tilde{\Theta}_i^1)^\mathsf{T}\Phi_i^1(y_i, \bar{Y}) + \tilde{\Theta}_{i,r,l}^2\Phi_{i,r,l}^2(y_i, \hat{y}_l^r) + \sum_{s \neq r}\tilde{\Theta}_{i,s}^2\Phi_{i,s}^2(y_i, \hat{Y}^s).$$

On the other hand, by Lemma 2, there exists some $p$, s.t., $\hat{y}_p^r \notin \mathcal{N}(y_j, \mathcal{Y}_r)$ (which by $\Phi_i^2 = \Phi_j^2$ further leads to $\hat{y}_p^r \notin \mathcal{N}(y_i, \mathcal{Y}_r)$). Setting $\hat{Y}_r = \hat{y}_l^r$ leads to

$$(\Theta_j^1)^\mathsf{T}\Phi_j^1(y_j, \bar{Y}) + (\tilde{\Theta}_i^1)^\mathsf{T}\Phi_i^1(y_j, \bar{Y}) + \sum_{s \neq r}\Theta_{j,s}^2\Phi_{j,s}^2(y_j, Y^s)$$
$$= (\Theta_j^1)^\mathsf{T}\Phi_j^1(y_i, \bar{Y}) + (\tilde{\Theta}_i^1)^\mathsf{T}\Phi_i^1(y_i, \bar{Y}) + \sum_{s \neq r}\tilde{\Theta}_{i,s}^2\Phi_{i,s}^2(y_i, \hat{Y}^s).$$

Subtracting the above two equations leads to $\Theta_{j,a,l}^2 = \tilde{\Theta}_{i,a,l}^2$. Since $a$ and $l$ are arbitrarily picked, we conclude that $\Theta_j^2 = \tilde{\Theta}_i^2$. Following the same routine, we also have $\Theta_i^2 = \tilde{\Theta}_j^2$.

Therefore, by applying $\Theta_j^1 = \tilde{\Theta}_i^1$, $\Theta_i^1 = \tilde{\Theta}_j^1$, $\Theta_j^2 = \tilde{\Theta}_i^2$, and $\Theta_i^2 = \tilde{\Theta}_j^2$ in Eq. (13), we have

$$(\Theta_i^1)^\mathsf{T}\Phi_i^1(y_i, \bar{Y}) - (\Theta_i^1)^\mathsf{T}\Phi_j^1(y_j, \bar{Y}) = (\Theta_i^1)^\mathsf{T}\Phi_i^1(y_j, \bar{Y}) - (\Theta_i^1)^\mathsf{T}\Phi_j^1(y_i, \bar{Y}),$$
$$(\Theta_j^1)^\mathsf{T}\Phi_j^1(y_j, \bar{Y}) - (\Theta_j^1)^\mathsf{T}\Phi_i^1(y_i, \bar{Y}) = (\Theta_j^1)^\mathsf{T}\Phi_j^1(y_i, \bar{Y}) - (\Theta_j^1)^\mathsf{T}\Phi_i^1(y_j, \bar{Y}).$$

Let $\bar{Y} = \mathbf{1}_{\hat{k}}$ (i.e., the $\hat{k}$-dimension all 1 vector), we have

$$(\Theta_i^1)^\mathsf{T}((\Phi_i^1(y_i, \mathbf{1}_{\hat{k}}) - \Phi_i^1(y_j, \mathbf{1}_{\hat{k}})) - ((\Phi_j^1(y_j, \mathbf{1}_{\hat{k}}) - \Phi_j^1(y_i, \mathbf{1}_{\hat{k}})))) = 0, \qquad (16)$$

$$(\Theta_j^1)^\mathsf{T}((\Phi_i^1(y_i, \mathbf{1}_{\hat{k}}) - \Phi_i^1(y_j, \mathbf{1}_{\hat{k}})) - ((\Phi_j^1(y_j, \mathbf{1}_{\hat{k}}) - \Phi_j^1(y_i, \mathbf{1}_{\hat{k}})))) = 0. \qquad (17)$$

Since $y_i$ and $y_j$ are asymmetric, we have that there exists $l$, such that $t_{y_i \hat{y}_l} \neq t_{y_j \hat{y}_l}$. Concretely, by Eq. (15), we have $t_{y_i \hat{y}_l} \in \{t^o, t^{sd}, t^{sg}\}$, $t_{y_j \hat{y}_l} = \{t^o, t^{sd}, t^{sg}\}$, and $t_{y_i \hat{y}_l} \neq t_{y_j \hat{y}_l}$. On the other hand,

$$\Phi_{i,l}^1(y_i, 1) - \Phi_{i,l}^1(y_j, 1)) = \Phi_{j,l}^1(y_j, 1) - \Phi_{j,l}^1(y_i, 1),$$

if and only if $t_{y_i \hat{y}_l} = t^o$, $t_{y_j \hat{y}_l} = t^{sg}$, or $t_{y_j \hat{y}_l} = t^o$, $t_{y_i \hat{y}_l} = t^{sg}$, which contradicts Lemma 1. On the other hand,

Therefore,

$$\Phi_{i,l}^1(y_i, 1) - \Phi_{i,l}^1(y_j, 1)) \neq \Phi_{j,l}^1(y_j, 1) - \Phi_{j,l}^1(y_i, 1).$$

In this case, solutions of $\Theta_i^1, \Theta_j^1$ subject to respectively Eqs. (16) and (17) lie along a zero-measure set.

The proof is completed. $\qquad \square$

# D    PROOF OF THEOREM 1

## D.1    LEARNING ALGORITHM

We first present the algorithm for producing $\hat{\Theta}$ and $\hat{W}$ in Algorithm 1.

---

**Algorithm 1** WIS

---

**Require:** Step size $\eta$, dataset $D \subset \mathcal{X}$, and initial parameter $\Theta_0$.

$\quad \hat{\Theta} \rightarrow \Theta_0$.

$\quad$ **for all** $X \in D$ **do**

$\qquad$ Independently sample $(Y, \bar{Y}, \hat{Y})$ from $\pi_{\hat{\Theta}}$, and $(Y', \bar{Y}', \hat{Y}')$ from $\pi_{\hat{\Theta}}$ conditionally given $\hat{Y}' = \hat{Y}(X)$.

$\qquad \hat{\Theta} \leftarrow \hat{\Theta} + \eta(\Phi(Y, \bar{Y}, \hat{Y}) - \Phi(Y', \bar{Y}', \hat{Y}'))$.

$\qquad$ Compute $\hat{W}$ as described in (3) using $\hat{\Theta}$.

$\quad$ **output** $(\hat{\Theta}, \hat{W})$

---

## D.2    ASSUMPTIONS

First, the problem distribution $\pi^*$ needs to be accurately modeled by some distribution $\Theta^*$ in the family that we are trying to learn:

$$\exists \Theta^* \text{ s.t. } \forall (Y, \hat{Y}), \, p_{(X,Y) \sim \pi^*}(Y, \hat{Y}) = p_{\theta^*}(Y, \hat{Y}). \qquad (18)$$

Secondly, given an example $(X, Y) \sim \pi^*$, we assume $Y$ is independent of $X$ given $\hat{Y}(X)$:

$$(X, Y) \sim \pi^* \Rightarrow Y \perp X \mid \hat{Y}(X). \qquad (19)$$

This assumption encodes the idea that while the ILFs can be arbitrarily dependent on the features, they provide sufficient information to accurately identify the true label vector. Then, for any $\Theta$, accurately learning $\Theta$ from data distribution is possible. That is, there exists an unbiased estimator $\hat{\Theta}(D)$ which is a function of the dataset $D$ of i.i.d from $\pi_\Theta$, such that, for any $\Theta$ and some $c > 0$,

$$\mathbf{Cov}(\hat{\Theta}(D)) \preceq \frac{I}{2c|D|}. \qquad (20)$$

And we are reasonably certain in our guess of latent variables, *i.e.*, $Y$ and $\bar{Y}$. That is, for any $\Theta, \Theta^*$,

$$\mathbb{E}_{\hat{Y}^* \sim \Theta^*} \left[ \sum_{i=1}^{k} (n_i + \hat{k}) \mathbf{Var}_{(Y, \bar{Y}, \hat{Y}) \sim \pi_\Theta} (\mathbb{1}_{Y=y_i} | \hat{Y} = \hat{Y}^*)^2 + \sum_{i=1}^{\hat{k}} (m_i + K - 1) \mathbf{Var}_{(Y, \bar{Y}, \hat{Y}) \sim \pi_\Theta} (\bar{Y}^i | \hat{Y} = \hat{Y}^*)^2 \right]^{\frac{1}{2}}$$

$$\leq \frac{c}{\sqrt{2M}}. \qquad (21)$$

We also assume that the output of the last layer of end model $h_W$ has bounded $\ell_\infty$ norm, that is, for any possible parameter $W$,

$$\|h_W\|_\infty \leq H. \tag{22}$$

Finally, we assume that solving Eq. (3) has bounded generalization risk such that for some $\chi > 0$, solution $\hat{W}$ satisfies

$$\mathbb{E}_{\hat{W}}\left[\ell_{\hat{\Theta}}(\hat{W}) - \min_W \ell_{\hat{\Theta}}(W)\right] \leq \chi. \tag{23}$$

### D.3 PROOF OF THEOREM 1

To begin with, we state two basic lemmas needed for proofs throughout this section:

**Lemma D.1.** *Let* $\mathbf{x}_1$, $\mathbf{x}_2$ *be two binary random variable. Then we have variance of product of* $\mathbf{x}_1$ *and* $\mathbf{x}_2$ *can be bounded as*

$$\mathbf{Var}\left[\mathbf{x}_1\mathbf{x}_2\right] \leq \mathbf{Var}\left[\mathbf{x}_1\right] + \mathbf{Var}\left[\mathbf{x}_2\right].$$

**Lemma D.2.** *Let* $Y$ *be a random vector and* $\|\cdot\|_s$ *be the spectral norm. Then we have*

$$\|\mathbf{Cov}(Y,Y)\|_s \leq \sum_i \mathbf{Var}(Y_i).$$

Then, we borrow two lemmas from (Ratner et al., 2016), which are slightly different from the original ones but can be easily proved following the same derivations:

**Lemma D.3.** *[Lemma D.1 in (Ratner et al., 2016)] Given a family of maximum-entropy distributions*

$$\pi_\Theta(Y, \bar{Y}, \hat{Y}) = \frac{1}{Z_\Theta} \exp\left(\Theta^\mathsf{T} \Phi(Y, \bar{Y}, \hat{Y})\right).$$

*If we let* $J$ *be the maximum expected log-likelihood objective, under another distribution* $\pi^*$, *for the event associated with the observed labeling function values* $\hat{Y}$,

$$J(\Theta) = \mathbb{E}_{\left(Y^*, \bar{Y}^*, \hat{Y}^*\right) \sim \pi^*}\left[\log \mathbb{P}_{(Y, \bar{Y}, \hat{Y}) \sim \pi_\Theta}\left(\hat{Y} = \hat{Y}^*\right)\right],$$

*then its Hessian can be calculated as*

$$\nabla^2 J(\Theta) = \mathbb{E}_{\left(Y^*, \bar{Y}^*, \hat{Y}^*\right) \sim \pi^*}\left[\mathbf{Cov}_{(Y, \bar{Y}, \hat{Y}) \sim \pi_\Theta}\left(\phi(Y, \bar{Y}, \hat{Y}) \mid \hat{Y} = \hat{Y}^*\right)\right] - \mathbf{Cov}_{(Y, \bar{Y}, \hat{Y}) \sim \pi_\Theta}(\phi(Y, \bar{Y}, \hat{Y})).$$

**Lemma D.4.** *[Lemma D.4 in (Ratner et al., 2016)] Suppose that we are looking at a WIS maximum likelihood estimation problem and the objective function* $J(\Theta)$ *is strongly concave with concavity parameter* $c > 0$. *If we run stochastic gradient descent using unbiased samples from a true distribution* $\pi_{\Theta^*}$, *then if we set step size as*

$$\eta = \frac{c\epsilon^2}{4},$$

*and run (using a fresh sample at each iteration) for* $T$ *steps, where*

$$T = \frac{2}{c^2\epsilon^2} \log\left(\frac{2\|\Theta_0 - \Theta^*\|^2}{\epsilon}\right).$$

*We can bound the expected parameter estimation error with*

$$\mathbb{E}\left\|\hat{\Theta} - \Theta^*\right\|^2 \leq M\epsilon^2, \tag{24}$$

*where* $M$ *is the dimension of* $\Theta$.

Based on Lemma D.4, in order to obtain the optimization error with respect to the estimated $\hat{\Theta}$ produced by Algorithm 1, we only need to show that the WIS object function $J(\Theta)^2$ is strongly concave. We prove this through the following lemma, which is a non-trivial extension of Lemma D.3 in (Ratner et al., 2016) given the fact that we have multiple latent variables and relatively complex dependency structures with comparison to (Ratner et al., 2016):

---

[2]Note that, in the Eq. (2) of the main body of the paper, we are minimizing $-J(\Theta)$, which is equivalent to maximizing $J(\Theta)$ as discussed here.

**Lemma D.5.** *[Extension of Lemma D.3 in (Ratner et al., 2016)] With conditions (20) and (21), the WIS objective function $J(\Theta)$ is strongly concave with strong convexity c.*

We then come to bound the generalization error of $\hat{W}$ produced by Algorithm 1, using the following non-trivial extension of Lemma D.5 in (Ratner et al., 2016):

**Lemma D.6.** *[Extension of Lemma D.5 in (Ratner et al., 2016)] Suppose that conditions (18)-(23) hold. Let $\hat{W}$ be the learned parameters of the end model produced by Algorithm 1, and $\ell(W^*)$ be the minimum of cross entropy loss function $\ell$. Then, we can bound the expected risk with*

$$\mathbb{E}\left[\ell(\hat{W}) - \ell(W^*)\right] \leq \chi + 4cH\epsilon.$$

Finally, we conclude Lemmas (D.4), (D.5) and (D.6) as the following theorem, which is identical to the Theorem 1 in the main body of the paper:

**Theorem 4** (Extension of Theorem 2 in (Ratner et al., 2016)). *Suppose that we run Algoirthm 1 on a WIS specification to produce $\hat{\Theta}$ and $\hat{W}$, and all conditions of Lemmas (D.5) and (D.6) are satisfied. Then, for any $\epsilon > 0$, if we set the step size to be*

$$\eta = \frac{c\epsilon^2}{4}$$

*and the input dataset $D$ is large enough such that*

$$|D| > \frac{2}{c^2\epsilon^2} \log\left(\frac{2\|\Theta_0 - \Theta^*\|^2}{\epsilon}\right),$$

*then we can bound the expected parameter error and the expected risk as:*

$$\mathbb{E}\left\|\hat{\Theta} - \Theta^*\right\|^2 \leq M\epsilon^2, \quad \mathbb{E}\left[\ell(\hat{W}) - \ell(W^*)\right] \leq \chi + 4cH\epsilon.$$

## D.4 PROOFS OF LEMMAS

**Lemma D.1.** *Let $\mathbf{x}_1$, $\mathbf{x}_2$ be two binary random variable. Then we have variance of product of $\mathbf{x}_1$ and $\mathbf{x}_2$ can be bounded as*

$$\mathbf{Var}\left[\mathbf{x}_1\mathbf{x}_2\right] \leq \mathbf{Var}\left[\mathbf{x}_1\right] + \mathbf{Var}\left[\mathbf{x}_2\right].$$

*Proof.* Joint distribution of $\mathbf{x}_1$ and $\mathbf{x}_2$ can be listed as the following table: (where $p_1 + p_2 + p_3 + p_4 = 1$)

| $\mathbf{x_1}/\mathbf{x_2}$ | 0 | 1 |
|:---:|:---:|:---:|
| 0 | $p_1$ | $p_2$ |
| 1 | $p_3$ | $p_4$ |

Then we have

$$\mathbf{Var}\left[\mathbf{x}_1\mathbf{x}_2\right] = p_4 - p_4^2 = p_4(p_1 + p_2 + p_3),$$

while

$$\mathbf{Var}\left[\mathbf{X}_1\right] + \mathbf{Var}\left[\mathbf{X}_2\right] = (p_2 + p_4)(p_1 + p_3) + (p_3 + p_4)(p_1 + p_2) \geq p_4(p_1 + p_2 + p_3).$$

The proof is completed. $\square$

**Lemma D.2.** *Let $Y$ be a random vector and $\|\cdot\|_s$ be the spectral norm. Then we have*

$$\|\mathbf{Cov}(Y, Y)\|_s \leq \sum_i \mathbf{Var}(Y_i).$$

*Proof.* By definition of spectral norm, we have

$$\|\mathbf{Cov}(Y,Y)\|_s = \max_{\|\boldsymbol{x}\|_2 \leq 1} \boldsymbol{x}^\mathsf{T} \mathbf{Cov}(Y,Y) \boldsymbol{x}$$

Where $\boldsymbol{x}$ is a constant vector. And by Cauchy-Schwarz inequality,

$$\boldsymbol{x}^\mathsf{T} \mathbf{Cov}(Y,Y) \boldsymbol{x} = \mathbb{E}\left[\boldsymbol{x}^\mathsf{T}(Y - \mathbb{E}[Y])(Y - \mathbb{E}[Y])^\mathsf{T} \boldsymbol{x}\right] \leq \mathbb{E}\left[\|\boldsymbol{x}\|^2 \|Y - \mathbb{E}[Y]\|^2\right].$$

Because $\boldsymbol{x}$ is a constant vector and $\|\boldsymbol{x}\| \leq 1$,

$$\max_{\|\boldsymbol{x}\|_2 \leq 1} \mathbb{E}\left[\|\boldsymbol{x}\|^2 \|Y - \mathbb{E}[Y]\|^2\right]$$

$$= \max_{\|\boldsymbol{x}\|_2 \leq 1} \|\boldsymbol{x}\|^2 \mathbb{E}\left[\|Y - \mathbb{E}[Y]\|^2\right]$$

$$= \max_{\|\boldsymbol{x}\|_2 \leq 1} \|\boldsymbol{x}\|^2 \left[\sum_i \mathbf{Var}(Y_i)\right]$$

$$= \sum_i \mathbf{Var}(Y_i).$$

The proof is completed. $\square$

**Lemma D.5.** *[Extension of Lemma D.3 in (Ratner et al., 2016)] With conditions (20) and (21), the WIS objective function $J(\Theta)$ is strongly concave with strong convexity c.*

*Proof.* By Lemma D.3, hessian matrix of $J$ can be decomposed as follows:

$$\nabla^2 J(\Theta) = \mathbb{E}_{\hat{Y}^* \sim \pi_{\Theta^*}}\left[\mathbf{Cov}_{(Y,\bar{Y},\hat{Y}) \sim \pi_\Theta}\left(\Phi(Y,\bar{Y},\hat{Y}) \mid \hat{Y} = \hat{Y}^*\right)\right] - \mathbf{Cov}_{(Y,\bar{Y},\hat{Y}) \sim \pi_\Theta}(\Phi(Y,\bar{Y},\hat{Y})).$$

Basically, to prove that $J(\Theta)$ is strongly concave with strong convexity $c$, we need to show for a real number $c > 0$,

$$\nabla^2 J(\Theta) \preceq c\mathbf{I}.$$

We calculate each term separately: for the first term

$$A = \mathbb{E}_{\hat{Y}^* \sim \pi_{\Theta^*}}\left[\mathbf{Cov}_{(Y,\bar{Y},\hat{Y}) \sim \pi_\Theta}\left(\Phi(Y,\bar{Y},\hat{Y}) \mid \hat{Y} = \hat{Y}^*\right)\right],$$

since $A$ is symmetric, for any real number $c$, $A \preceq c\mathbf{I}$, if and only if its spectral norm $\|A\|_s \leq c$, where $\|A\|_s$ equals to the eigenvalue of $A$ with largest absolute value.

Since by definition, vector function $\Phi(Y,\bar{Y},\hat{Y})$ can be represented as:

$$\Phi(Y,\bar{Y},\hat{Y}) = \begin{pmatrix} \left(\phi^{\mathrm{Acc}}_{y_i,\hat{y}^j_l,j}(Y,\hat{Y}^j)\right)_{i \in [k], j \in [n], \hat{y}^j_l \in \mathcal{N}(y_i,\mathcal{Y}_j)} \\ \left(\phi^{\mathrm{Acc}}_{\hat{y}_i,j}(\bar{Y}^i,\hat{Y}^j)\right)_{j \in [n], \hat{y}_i \in \mathcal{Y}_j} \\ \left(\phi^t_{\hat{y}_i,\hat{y}_j}(\bar{Y}^i,\bar{Y}^j)\right)_{i,j \in [\hat{k}]} \\ \left(\phi^t_{y_i,\hat{y}_j}(Y,\bar{Y}^j)\right)_{i \in [k], j \in [\hat{k}]} \end{pmatrix},$$

by Lemma D.2, we have $A$ can be further bounded by

$$A \le \left( \mathbb{E}_{\hat{Y}^* \sim \pi_{\Theta^*}} \left[ \left( \sum_{i=1}^{k} \sum_{j=1}^{n} \sum_{\hat{y}_l^j \in \mathcal{N}(y_i, \mathcal{Y}_j)} \mathbf{Var}_{(Y,\bar{Y},\hat{Y}) \sim \pi_\Theta} \left( \phi^{\mathrm{Acc}}_{y_i, \hat{y}_l^j, j}(Y, \hat{Y}^j) \mid \hat{Y} = \hat{Y}^* \right) \right) \right] \right.$$

$$+ \mathbb{E}_{\hat{Y}^* \sim \pi_{\Theta^*}} \left[ \left( \sum_{j=1}^{n} \sum_{\hat{y}_i \in \mathcal{Y}_j} \mathbf{Var}_{(Y,\bar{Y},\hat{Y}) \sim \pi_\Theta} \left( \phi^{\mathrm{Acc}}_{\hat{y}_i, j}(\bar{Y}^i, \hat{Y}^j) \mid \hat{Y} = \hat{Y}^* \right) \right) \right]$$

$$+ \mathbb{E}_{\hat{Y}^* \sim \pi_{\Theta^*}} \left[ \left( \sum_{1 \le i,j \le \hat{k}} \mathbf{Var}_{(Y,\bar{Y},\hat{Y}) \sim \pi_\Theta} \left( \phi^{t}_{\hat{y}_i, \hat{y}_j}(\bar{Y}^i, \bar{Y}^j) \mid \hat{Y} = \hat{Y}^* \right) \right) \right]$$

$$\left. + \mathbb{E}_{\hat{Y}^* \sim \pi_{\Theta^*}} \left[ \left( \sum_{i=1}^{k} \sum_{j=1}^{\hat{k}} \mathbf{Var}_{(Y,\bar{Y},\hat{Y}) \sim \pi_\Theta} \left( \phi^{t}_{y_i, \hat{y}_j}(Y, \bar{Y}^j) \mid \hat{Y} = \hat{Y}^* \right) \right) \right] \right)$$

$$= A_1 + A_2 + A_3 + A_4,$$

where

$$A_1 = \mathbb{E}_{\hat{Y}^* \sim \pi_{\Theta^*}} \left[ \left( \sum_{i=1}^{k} \sum_{j=1}^{n} \sum_{\hat{y}_l^j \in \mathcal{N}(y_i, \mathcal{Y}_j)} \mathbf{Var}_{(Y,\bar{Y},\hat{Y}) \sim \pi_\Theta} \left( \phi^{\mathrm{Acc}}_{y_i, \hat{y}_l^j, j}(Y, \hat{Y}^j) \mid \hat{Y} = \hat{Y}^* \right) \right) \right];$$

$$A_2 = \mathbb{E}_{\hat{Y}^* \sim \pi_{\Theta^*}} \left[ \left( \sum_{j=1}^{n} \sum_{\hat{y}_l \in \mathcal{Y}_j} \mathbf{Var}_{(Y,\bar{Y},\hat{Y}) \sim \pi_\Theta} \left( \phi^{\mathrm{Acc}}_{\hat{y}_l, j}(Y, \hat{Y}^j) \mid \hat{Y} = \hat{Y}^* \right) \right) \right];$$

$$A_3 = \mathbb{E}_{\hat{Y}^* \sim \pi_{\Theta^*}} \left[ \left( \sum_{1 \le i,j \le \hat{k}} \mathbf{Var}_{(Y,\bar{Y},\hat{Y}) \sim \pi_\Theta} \left( \phi^{t}_{\hat{y}_i, \hat{y}_j}(\bar{Y}^i, \bar{Y}^j) \mid \hat{Y} = \hat{Y}^* \right) \right) \right];$$

$$A_4 = \mathbb{E}_{\hat{Y}^* \sim \pi_{\Theta^*}} \left[ \left( \sum_{i=1}^{k} \sum_{j=1}^{\hat{k}} \mathbf{Var}_{(Y,\bar{Y},\hat{Y}) \sim \pi_\Theta} \left( \phi^{t}_{y_i, \hat{y}_j}(Y, \bar{Y}^j) \mid \hat{Y} = \hat{Y}^* \right) \right) \right].$$

We then bound the four terms respectively. As for $A_1$, for fixed $\hat{Y}^*$, we have

$$\sum_{i=1}^{k} \sum_{j=1}^{n} \sum_{\hat{y}_l^j \in \mathcal{N}(y_i, \mathcal{Y}_j)} \mathbf{Var}_{(Y,\bar{Y},\hat{Y}) \sim \pi_\Theta} \left( \phi^{\mathrm{Acc}}_{y_i, \hat{y}_l^j, j}(Y, \hat{Y}^j) \mid \hat{Y} = \hat{Y}^* \right)$$

$$= \sum_{i=1}^{k} \sum_{j=1}^{n} \sum_{\hat{y}_l^j \in \mathcal{N}(y_i, \mathcal{Y}_j)} \mathbf{Var}_{(Y,\bar{Y},\hat{Y}) \sim \pi_\Theta} \left( \mathbb{1}_{Y = y_i \wedge \hat{Y}^j = \hat{y}_l^j} \mid \hat{Y} = \hat{Y}^* \right)$$

$$= \sum_{i=1}^{k} \left[ \sum_{j \in [n], \hat{y}_l^j \in \mathcal{N}(y_i, \mathcal{Y}_j), (\hat{Y}^*)^j = \hat{y}_l^j} \mathbf{Var}_{(Y,\bar{Y},\hat{Y}) \sim \pi_\Theta} \left( \mathbb{1}_{Y = y_i} \mid \hat{Y} = \hat{Y}^* \right) \right]$$

$$= \sum_{i=1}^{k} \left[ \sum_{j \in [n], \hat{y}_l^j \in \mathcal{N}(y_i, \mathcal{Y}_j), (\hat{Y}^*)^j = \hat{y}_l^j} \mathbf{Var}_{(Y,\bar{Y},\hat{Y}) \sim \pi_\Theta} \left( \mathbb{1}_{Y = y_i} \mid \hat{Y} = \hat{Y}^* \right) \right]$$

$$\le \sum_{i=1}^{k} n_i \mathbf{Var}_{(Y,\bar{Y},\hat{Y}) \sim \pi_\Theta} \left( \mathbb{1}_{Y = y_i} \mid \hat{Y} = \hat{Y}^* \right),$$

where $n_i$ is the number of ILFs whose label space contains label that is non-exclusive to label $y_i$, *i.e.*, $n_i = |\{ j \in [n] | \mathcal{N}(y_i, \mathcal{Y}_j) \ne \emptyset \}|$.

Therefore, we have

$$A_1 \le \sum_{i=1}^{k} n_i \mathbb{E}_{\hat{Y}^* \sim \pi_{\Theta^*}} \mathbf{Var}_{(Y,\bar{Y},\hat{Y}) \sim \pi_\Theta} \left( \mathbb{1}_{Y = y_i} \mid \hat{Y} = \hat{Y}^* \right).$$

Similarly, for $A_2$, we have

$$A_2 \le \sum_{i=1}^{\hat{k}} m_i \mathbb{E}_{\hat{Y}^* \sim \pi_{\Theta^*}} \mathbf{Var}_{(Y,\bar{Y},\hat{Y}) \sim \pi_{\Theta}} \left( \bar{Y}^i \mid \hat{Y} = \hat{Y}^* \right),$$

where $m_i$ is the number of ILFs whose label space contains the label $\hat{y}_i$.

As for $A_3$, for fixed $\hat{Y}^*$ and any $\hat{y}_i, \hat{y}_j \in \hat{\mathcal{Y}}$, we further separate the proof into subcases by $t_{\hat{y}_i \hat{y}_j}$ which is simplified as $t$:

(1). $t = t^o$. In this case,

$$\mathbf{Var}_{(Y,\bar{Y},\hat{Y}) \sim \pi_{\Theta}} \left( \phi_{\hat{y}_i, \hat{y}_j}^t (\bar{Y}^i, \bar{Y}^j) \mid \hat{Y} = \hat{Y}^* \right)$$
$$= \mathbf{Var}_{(Y,\bar{Y},\hat{Y}) \sim \pi_{\Theta}} \left( \mathbf{1}_{\bar{Y}^i = \bar{Y}^j} \mid \hat{Y} = \hat{Y}^* \right)$$
$$= \mathbf{Var}_{(Y,\bar{Y},\hat{Y}) \sim \pi_{\Theta}} \left( \bar{Y}^i \bar{Y}^j \mid \hat{Y} = \hat{Y}^* \right)$$
$$\overset{(*)}{\le} \mathbf{Var}_{(Y,\bar{Y},\hat{Y}) \sim \pi_{\Theta}} \left( \bar{Y}^i \mid \hat{Y} = \hat{Y}^* \right) + \mathbf{Var}_{(Y,\bar{Y},\hat{Y}) \sim \pi_{\Theta}} \left( \bar{Y}^j \mid \hat{Y} = \hat{Y}^* \right),$$

where Eq. $(*)$ is due to Lemma D.1.

(2). $t = t^e$. Similarly,

$$\mathbf{Var}_{(Y,\bar{Y},\hat{Y}) \sim \pi_{\Theta}} \left( \phi_{\hat{y}_i, \hat{y}_j}^t (\bar{Y}^i, \bar{Y}^j) \mid \hat{Y} = \hat{Y}^* \right)$$
$$= \mathbf{Var}_{(Y,\bar{Y},\hat{Y}) \sim \pi_{\Theta}} \left( -\mathbf{1}_{\bar{Y}^i = \bar{Y}^j = 1} \mid \hat{Y} = \hat{Y}^* \right)$$
$$= \mathbf{Var}_{(Y,\bar{Y},\hat{Y}) \sim \pi_{\Theta}} \left( \mathbf{1}_{\bar{Y}^i = \bar{Y}^j = 1} \mid \hat{Y} = \hat{Y}^* \right)$$
$$= \mathbf{Var}_{(Y,\bar{Y},\hat{Y}) \sim \pi_{\Theta}} \left( \bar{Y}^i \bar{Y}^j \mid \hat{Y} = \hat{Y}^* \right)$$
$$\le \mathbf{Var}_{(Y,\bar{Y},\hat{Y}) \sim \pi_{\Theta}} \left( \bar{Y}^i \mid \hat{Y} = \hat{Y}^* \right) + \mathbf{Var}_{(Y,\bar{Y},\hat{Y}) \sim \pi_{\Theta}} \left( \bar{Y}^j \mid \hat{Y} = \hat{Y}^* \right),$$

(3). $t = t^{sg}$. In this case,

$$\mathbf{Var}_{(Y,\bar{Y},\hat{Y}) \sim \pi_{\Theta}} \left( \phi_{\hat{y}_i, \hat{y}_j}^t (\bar{Y}^i, \bar{Y}^j) \mid \hat{Y} = \hat{Y}^* \right)$$
$$= \mathbf{Var}_{(Y,\bar{Y},\hat{Y}) \sim \pi_{\Theta}} \left( -\mathbf{1}_{\bar{Y}^i = 1, \bar{Y}^j = 0} \mid \hat{Y} = \hat{Y}^* \right)$$
$$= \mathbf{Var}_{(Y,\bar{Y},\hat{Y}) \sim \pi_{\Theta}} \left( (1 - \bar{Y}^i) \bar{Y}^j \mid \hat{Y} = \hat{Y}^* \right)$$
$$\le \mathbf{Var}_{(Y,\bar{Y},\hat{Y}) \sim \pi_{\Theta}} \left( 1 - \bar{Y}^i \mid \hat{Y} = \hat{Y}^* \right) + \mathbf{Var}_{(Y,\bar{Y},\hat{Y}) \sim \pi_{\Theta}} \left( \bar{Y}^j \mid \hat{Y} = \hat{Y}^* \right)$$
$$= \mathbf{Var}_{(Y,\bar{Y},\hat{Y}) \sim \pi_{\Theta}} \left( \bar{Y}^i \mid \hat{Y} = \hat{Y}^* \right) + \mathbf{Var}_{(Y,\bar{Y},\hat{Y}) \sim \pi_{\Theta}} \left( \bar{Y}^j \mid \hat{Y} = \hat{Y}^* \right),$$

(4). $t = t^{sd}$. Similar to (3).,

$$\mathbf{Var}_{(Y,\bar{Y},\hat{Y}) \sim \pi_{\Theta}} \left( \phi_{\hat{y}_i, \hat{y}_j}^t (\bar{Y}^i, \bar{Y}^j) \mid \hat{Y} = \hat{Y}^* \right)$$
$$= \mathbf{Var}_{(Y,\bar{Y},\hat{Y}) \sim \pi_{\Theta}} \left( -\mathbf{1}_{\bar{Y}^i = 0, \bar{Y}^j = 1} \mid \hat{Y} = \hat{Y}^* \right)$$
$$= \mathbf{Var}_{(Y,\bar{Y},\hat{Y}) \sim \pi_{\Theta}} \left( (1 - \bar{Y}^j) \bar{Y}^i \mid \hat{Y} = \hat{Y}^* \right)$$
$$\le \mathbf{Var}_{(Y,\bar{Y},\hat{Y}) \sim \pi_{\Theta}} \left( 1 - \bar{Y}^j \mid \hat{Y} = \hat{Y}^* \right) + \mathbf{Var}_{(Y,\bar{Y},\hat{Y}) \sim \pi_{\Theta}} \left( \bar{Y}^i \mid \hat{Y} = \hat{Y}^* \right)$$
$$= \mathbf{Var}_{(Y,\bar{Y},\hat{Y}) \sim \pi_{\Theta}} \left( \bar{Y}^i \mid \hat{Y} = \hat{Y}^* \right) + \mathbf{Var}_{(Y,\bar{Y},\hat{Y}) \sim \pi_{\Theta}} \left( \bar{Y}^j \mid \hat{Y} = \hat{Y}^* \right),$$

Combining (1), (2), (3), and (4), we have

$$A_3 \leq \sum_{i=1}^{\hat{k}} (\hat{k}-1) \mathbb{E}_{\hat{Y}^* \sim \pi_{\Theta^*}} \mathbf{Var}_{(Y,\bar{Y},\hat{Y}) \sim \pi_\Theta} \left( \bar{Y}^i \mid \hat{Y} = \hat{Y}^* \right),$$

As for $A_4$, by similar discussion of $A_3$,

$$A_4 \leq \sum_{i=1}^{\hat{k}} k \mathbb{E}_{\hat{Y}^* \sim \pi_{\Theta^*}} \mathbf{Var}_{(Y,\bar{Y},\hat{Y}) \sim \pi_\Theta} \left( \bar{Y}^i \mid \hat{Y} = \hat{Y}^* \right) + \sum_{i=1}^{k} \hat{k} \mathbb{E}_{\hat{Y}^* \sim \pi_{\Theta^*}} \mathbf{Var}_{(Y,\bar{Y},\hat{Y}) \sim \pi_\Theta} \left( \mathbb{1}_{Y=y_i} \mid \hat{Y} = \hat{Y}^* \right).$$

Combining estimation of $A_1$, $A_2$, $A_3$, $A_4$, and by condition (21) we have

$$\|A\|_s$$
$$\leq A_1 + A_2 + A_3 + A_4$$
$$\leq \mathbb{E}_{\hat{Y}^* \sim \pi_{\Theta^*}} \left[ \sum_{i=1}^{k} (n_i + \hat{k}) \mathbf{Var}_{Y,\hat{Y}} (\mathbb{1}_{Y=y_i} | \hat{Y} = \hat{Y}^*) + \sum_{i=1}^{\hat{k}} (m_i + K - 1) \mathbf{Var}_{Y,\hat{Y}} (\bar{Y}^i | \hat{Y} = \hat{Y}^*) \right]$$
$$\leq \mathbb{E}_{\hat{Y}^* \sim \pi_{\Theta^*}} \left[ \sum_{i=1}^{k} (n_i + \hat{k}) \mathbf{Var}_{Y,\hat{Y}}^2 (Y | \hat{Y} = \hat{Y}^*) + \sum_{i=1}^{\hat{k}} (m_i + K - 1) \mathbf{Var}_{Y,\hat{Y}}^2 (\bar{Y}^i | \hat{Y} = \hat{Y}^*) \right]^{\frac{1}{2}}$$
$$\cdot \left( \sum_{i=1}^{k} (n_i + \hat{k}) + \sum_{i=1}^{K} (m_i + K - 1) \right)^{\frac{1}{2}}$$
$$\leq \frac{c}{\sqrt{2M}} \cdot \sqrt{2M} \leq c,$$

which further leads to

$$A \preceq c\mathbf{I}.$$

For the second term $B = \mathbf{Cov}_{(Y,\bar{Y},\hat{Y}) \sim \pi_\Theta} (\Phi(Y,\bar{Y},\hat{Y}))$,

$$B = \mathbb{E}_{(Y,\bar{Y},\hat{Y}) \sim \pi_\Theta} \left[ (\Phi(Y,\bar{Y},\hat{Y}) - \mathbb{E}_{(Y,\bar{Y},\hat{Y}) \sim \pi_\Theta} [\Phi(Y,\bar{Y},\hat{Y})])^2 \right]$$

$$= \mathbb{E}_{Y,\bar{Y},\hat{Y} \sim \pi_\Theta} \left[ \left( \Phi(Y,\bar{Y},\hat{Y}) - \frac{\sum_{Y',\bar{Y}',\hat{Y}'} \Phi(Y',\bar{Y}',\hat{Y}') \exp\left(\Theta^T \Phi(Y',\bar{Y}',\hat{Y}')\right)}{\sum_{Y',\bar{Y}',\hat{Y}'} \exp\left(\Theta^T \Phi(Y',\bar{Y}',\hat{Y}')\right)} \right)^2 \right]$$

$$= \mathbb{E}_{Y,\bar{Y},\hat{Y} \sim \pi_\Theta} \left[ \left( \nabla_\Theta \log \left( \exp\left(\Theta^T \Phi(Y,\bar{Y},\hat{Y})\right) \right) - \nabla_\Theta \log \left( \sum_{Y',\bar{Y}',\hat{Y}'} \exp\left(\Theta^T \Phi(Y',\bar{Y}',\hat{Y}')\right) \right) \right)^2 \right]$$

$$= \mathbb{E}_{(Y,\bar{Y},\hat{Y}) \sim \pi_\Theta} \left[ \left( \nabla_\Theta \log \pi_\Theta(Y,\bar{Y},\hat{Y}) \right)^2 \right],$$

where $\mathbb{E}_{(Y,\bar{Y},\hat{Y}) \sim \pi_\Theta} \left[ \left( \nabla_\Theta \log \pi_\Theta(Y,\bar{Y},\hat{Y}) \right)^2 \right]$ is the Fisher Information of $\Theta$. By the Cramér-Rao bound and the condition (20),

$$\frac{I}{2c|D|} \succeq \mathbf{Cov}(\hat{\Theta}) \succeq \left( D \mathbb{E}_{(Y,\hat{Y}) \sim \pi_\Theta} \left[ \left( \nabla_\Theta \log \pi_\Theta(Y,\hat{Y}) \right)^2 \right] \right)^{-1},$$

which further leads to

$$B = \mathbb{E}_{(Y,\bar{Y},\hat{Y}) \sim \pi_\Theta} \left[ \left( \nabla_\Theta \log \pi_\Theta(Y,\bar{Y},\hat{Y}) \right)^2 \right] \succeq 2c I.$$

The proof is completed by putting estimation of terms $A$ and $B$ together. □

**Lemma D.6.** *[Extension of Lemma D.5 in (Ratner et al., 2016)] Suppose that conditions (18)-(23) hold. Let $\hat{W}$ be the learned parameters of the end model produced by Algorithm 1, and $\ell(W^*)$ be the minimum of cross entropy loss function $\ell$. Then, we can bound the expected risk with*

$$\mathbb{E}\left[\ell(\hat{W}) - \ell(W^*)\right] \leq \chi + 4cH\epsilon.$$

*Proof.* We begin by rewriting objective of expected loss minimization problem using law of total expectation as follows:

$$\begin{aligned}
\ell(W) =& \mathbb{E}_{(X,Y)\sim\pi^*}\left[\mathbb{E}_{(X,Y)\sim\pi^*}\left[\mathcal{H}(Y, \sigma(h(X, W)))|X\right]\right] \\
=& \mathbb{E}_{(X',Y')\sim\pi^*}\left[\mathbb{E}_{(X,Y)\sim\pi^*}\left[\mathcal{H}(Y, \sigma(h(X, W)))|X = X'\right]\right] \\
=& \mathbb{E}_{(X',Y')\sim\pi^*}\left[\mathbb{E}_{(X,Y)\sim\pi^*}\left[\mathcal{H}(Y, \sigma(h(X', W)))|X = X'\right]\right]
\end{aligned}$$

and by our conditional independence assumption (condition (19)), we have

$$\mathbb{P}(Y|X = X') = \mathbb{P}(Y|\hat{Y}(X) = \hat{Y}(X')),$$

which further leads to

$$\begin{aligned}
\ell(W) =& \mathbb{E}_{(X',Y')\sim\pi^*}\left[\mathbb{E}_{(X,Y)\sim\pi^*}\left[\mathcal{H}(Y, \sigma(h(X', W)))\Big|\hat{Y}(X) = \hat{Y}(X')\right]\right] \\
=& \mathbb{E}_{(X',Y')\sim\pi^*}\left[\mathbb{E}_{(Y,\hat{Y})\sim\pi_{\Theta^*}}\left[\mathcal{H}(Y, \sigma(h(X', W)))\Big|\hat{Y} = \hat{Y}(X')\right]\right]
\end{aligned}$$

On the other hand, if we are minimizing the model with learned parameter $\hat{\Theta}$, we will be actually minimizing

$$\ell_{\hat{\Theta}}(W) = \mathbb{E}_{(X',Y')\sim\pi^*}\left[\mathbb{E}_{(Y,\hat{Y})\sim\pi_{\hat{\Theta}}}\left[\mathcal{H}(Y, \sigma(h(X', W)))\Big|\hat{Y} = \hat{Y}(X')\right]\right],$$

where for any $X'$, $\mathbb{E}_{(Y,\hat{Y})\sim\pi_{\hat{\Theta}}}\left[\mathcal{H}(Y, \sigma(h(X', W)))\Big|\hat{Y} = \hat{Y}(X')\right]$ can be further calculated as

$$\begin{aligned}
&\mathbb{E}_{(Y,\hat{Y})\sim\pi_{\hat{\Theta}}}\left[\mathcal{H}(Y, \sigma(h(X', W)))\Big|\hat{Y} = \hat{Y}(X')\right] \\
&= \sum_{l=1}^{k} \log\left(\sigma(h(X', W))_l\right)\mathbb{P}_{(Y,\hat{Y})\sim\pi_{\hat{\Theta}}}(Y = y_l|\hat{Y} = \hat{Y}(X')).
\end{aligned}$$

For simplification, we rewrite $\mathbb{P}_{(Y,\hat{Y})\sim\pi_{\hat{\Theta}}}(Y = y_l|\hat{Y} = \hat{Y}(X'))$ as follows with slight abuse of notations:

$$\mathbb{P}_{(Y,\hat{Y})\sim\pi_{\hat{\Theta}}}(Y = y_l|\hat{Y} = \hat{Y}(X')) = \mathbb{P}_{\pi_{\hat{\Theta}}}(y_l|\hat{Y}(X')),$$

and similarly

$$\mathbb{E}_{(X',Y')\sim\pi^*} = \mathbb{E}_{\pi^*},$$

Let $l_{X'} \triangleq \arg\min_l \log\left(\sigma(h(X', W))_l\right)$. The difference between the loss functions will be

$$\begin{aligned}
|\ell_{\hat{\Theta}}(W) - \ell(W)| =& \left|\mathbb{E}_{\pi^*}\left[\sum_{l=1}^{k}\log\left(\sigma(h(X', W))_l\right)\left(\mathbb{P}_{\pi_{\Theta^*}}(y_l|\hat{Y}(X')) - \mathbb{P}_{\pi_{\hat{\Theta}}}(y_l|\hat{Y}(X'))\right)\right]\right| \\
=& \left|\mathbb{E}_{\pi^*}\left[\log\left(\sigma(h(X', W))_{l_{X'}}\right)\left(\mathbb{P}_{\pi_{\Theta^*}}(y_{l_{X'}}|\hat{Y}(X')) - \mathbb{P}_{\pi_{\hat{\Theta}}}(y_{l_{X'}}|\hat{Y}(X'))\right)\right]\right. \\
& + \mathbb{E}_{\pi^*}\left[\sum_{l \neq l_{X'}}\log\left(\sigma(h(X', W))_l\right)\left(\mathbb{P}_{\pi_{\Theta^*}}(y_l|\hat{Y}(X')) - \mathbb{P}_{\pi_{\hat{\Theta}}}(y_l|\hat{Y}(X'))\right)\right]\right|.
\end{aligned}$$

Furthermore,

$$
\left| \mathbb{E}_{\pi^*} \left[ \log \left( \sigma(h(X', W))_{l_{X'}} \right) \left( \mathbb{P}_{\pi_{\Theta^*}}(y_{l_{X'}}|\hat{Y}(X')) - \mathbb{P}_{\pi_{\hat{\Theta}}}(y_{l_{X'}}|\hat{Y}(X')) \right) \right] \right.
$$

$$
\left. + \mathbb{E}_{\pi^*} \left[ \sum_{l \neq l_{X'}} \log \left( \sigma(h(X', W))_l \right) \left( \mathbb{P}_{\pi_{\Theta^*}}(y_l|\hat{Y}(X')) - \mathbb{P}_{\pi_{\hat{\Theta}}}(y_l|\hat{Y}(X')) \right) \right] \right|
$$

$$
= \left| \mathbb{E}_{\pi^*} \left[ \log \left( \sigma(h(X', W))_{l_{X'}} \right) \left( - \sum_{l \neq l_{X'}} \mathbb{P}_{\pi_{\Theta^*}}(y_l|\hat{Y}(X')) + \sum_{j \neq l_{X'}} \mathbb{P}_{\pi_{\hat{\Theta}}}(y_l|\hat{Y}(X')) \right) \right] \right.
$$

$$
\left. + \mathbb{E}_{\pi^*} \left[ \sum_{l \neq l_{X'}} \log \left( \sigma(h(X', W))_l \right) \left( \mathbb{P}_{\pi_{\Theta^*}}(y_l|\hat{Y}(X')) - \mathbb{P}_{\pi_{\hat{\Theta}}}(y_l|\hat{Y}(X')) \right) \right] \right|
$$

$$
= \left| \mathbb{E}_{\pi^*} \left[ \sum_{l \neq l_{X'}} \left( \log \left( \sigma(h(X', W))_l \right) - \log \left( \sigma(h(X', W))_{l_{X'}} \right) \right) \left( \mathbb{P}_{\pi_{\Theta^*}}(y_l|\hat{Y}(X')) - \mathbb{P}_{\pi_{\hat{\Theta}}}(y_l|\hat{Y}(X')) \right) \right] \right|
$$

$$
= \left| \mathbb{E}_{\pi^*} \left[ \sum_{l \neq l_{X'}} \left( h(X', W)_l - h(X', W)_{l_{X'}} \right) \left( \mathbb{P}_{\pi_{\Theta^*}}(y_l|\hat{Y}(X')). - \mathbb{P}_{\pi_{\hat{\Theta}}}(y_l|\hat{Y}(X')) \right) \right] \right|. \tag{25}
$$

Let

$$
\bar{h}(l_1, l_2) = h(X', W)_{l_1} - h(X', W)_{l_2}.
$$

By Eq. (22), we have for any $l \in [k]$,

$$
0 \leq \bar{h}(l, l_{X'}) \leq 2H.
$$

For any fixed $X'$, define $g_{X'}(\Theta)$ as follows:

$$
g_{X'}(\Theta) = \sum_{l \neq l_{X'}} \bar{h}(l, l_{X'}) \mathbb{P}_{\pi_\Theta}(y_l|\hat{Y}(X')), \tag{26}
$$

based on which we have

$$
\left| \ell_{\hat{\Theta}}(W) - \ell(W) \right| \leq \left| \mathbb{E}_{\pi^*} \left( g_{X'}(\hat{\Theta}) - g_{X'}(\Theta^*) \right) \right|
$$

By First Mean Value Theorem,

$$
g_{X'}(\hat{\Theta}) - g_X(\Theta^*) = \langle \nabla g_{X'}(\xi), \hat{\Theta} - \Theta^* \rangle \leq \|\hat{\Theta} - \Theta^*\| \|\nabla g_{X'}(\xi)\|.
$$

We then bound $\nabla g_X(\xi)$ element-wisely:

(1). For any $i \in [k], j \in [n], \hat{y}_l^j \in \mathcal{N}(y_i, \mathcal{Y}_j)$, if $i = l_{X'}, \hat{Y}^j(X') = \hat{y}_l^j$,

$$
\left| \frac{\partial g_{X'}(\xi)}{\partial \theta_{y_i, \hat{y}_l^j, j}^{\text{Acc}}} \right| = \left| \sum_{l \neq l_{X'}} \bar{h}(l, l_{X'}) \frac{\partial \mathbb{P}_{\pi_\xi}(y_l|\hat{Y}(X'))}{\partial \theta_{y_i, \hat{y}_l^j, j}^{\text{Acc}}} \right|
$$

$$
= \left| - \sum_{l \neq l_{X'}} \bar{h}(l, l_{X'}) \mathbb{P}_{\pi_\xi}(y_i|\hat{Y}(X')) \mathbb{P}_{\pi_\xi}(y_l|\hat{Y}(X')) \right|
$$

$$
= \sum_{l \neq l_{X'}} \bar{h}(l, l_{X'}) \mathbb{P}_{\pi_\xi}(y_i|\hat{Y}(X')) \mathbb{P}_{\pi_\xi}(y_l|\hat{Y}(X'))
$$

$$
\leq 2H \mathbb{P}_{\pi_\xi}(y_i|\hat{Y}(X'))(1 - \mathbb{P}_{\pi_\xi}(y_i|\hat{Y}(X')))
$$

$$
= 2H \mathbf{Var} \left[ \mathbb{1}_{Y=y_i}|\hat{Y}(X') \right].
$$

If $i \neq l_{X'}$, $\hat{Y}^j(X') = \hat{y}_l^j$,

$$
\begin{aligned}
\left| \frac{\partial g_{X'}(\xi)}{\partial \theta^{\text{Acc}}_{y_i, \hat{y}_l^j, j}} \right| &= \left| \sum_{l \neq l_{X'}} \bar{h}(l, l_{X'}) \frac{\partial \mathbb{P}_{\pi_\xi}(y_l | \hat{Y}(X'))}{\partial \theta^{\text{Acc}}_{y_i, \hat{y}_l^j, j}} \right| \\
&= \left| - \sum_{l \notin \{i, l_{X'}\}} \bar{h}(l, l_{X'}) \mathbb{P}_{\pi_\xi}(y_i | \hat{Y}(X')) \mathbb{P}_{\pi_\xi}(y_l | \hat{Y}(X')) \right. \\
&\quad \left. + \bar{h}(i, l_{X'}) \left( \mathbb{P}_{\pi_\xi}(y_i | \hat{Y}(X')) - \mathbb{P}_{\pi_\xi}(y_i | \hat{Y}(X')) \mathbb{P}_{\pi_\xi}(y_i | \hat{Y}(X')) \right) \right| \\
&\leq \max \left\{ \sum_{l \notin \{i, l_{X'}\}} \bar{h}(l, l_{X'}) \mathbb{P}_{\pi_\xi}(y_i | \hat{Y}(X')) \mathbb{P}_{\pi_\xi}(y_l | \hat{Y}(X')), \right. \\
&\quad \left. \bar{h}(i, l_{X'}) \left( \mathbb{P}_{\pi_\xi}(y_i | \hat{Y}(X')) - \mathbb{P}_{\pi_\xi}(y_i | \hat{Y}(X')) \mathbb{P}_{\pi_\xi}(y_i | \hat{Y}(X')) \right) \right\} \\
&\leq 2H \mathbb{P}_{\pi_\xi}(y_i | \hat{Y}(X')) (1 - \mathbb{P}_{\pi_\xi}(y_i | \hat{Y}(X'))) \\
&= 2H \mathbf{Var} \left[ \mathbb{1}_{Y = y_i} | \hat{Y}(X') \right].
\end{aligned}
$$

If $\hat{Y}^j(X') \neq \hat{y}_l^j$,

$$
\left| \frac{\partial g_{X'}(\xi)}{\partial \theta^{\text{Acc}}_{y_i, \hat{y}_l^j, j}} \right| = \left| \sum_{l \neq l_{X'}} \bar{h}(l, l_{X'}) \frac{\partial \mathbb{P}_{\pi_\xi}(y_l | \hat{Y}(X'))}{\partial \theta^{\text{Acc}}_{y_i, \hat{y}_l^j, j}} \right| = 0.
$$

(2). For $j \in [n], \hat{y}_r \in \mathcal{Y}_j$, if $\hat{Y}^j(X') = \hat{y}_r$,

$$
\begin{aligned}
\left| \frac{\partial g_{X'}(\xi)}{\partial \theta^{\text{Acc}}_{\hat{y}_r, j}} \right| &= \left| \sum_{l \neq l_{X'}} \bar{h}(l, l_{X'}) \frac{\partial \mathbb{P}_{\pi_\xi}(y_l | \hat{Y}(X'))}{\partial \theta^{\text{Acc}}_{\hat{y}_r, j}} \right| \\
&= \left| \sum_{l \neq l_{X'}} \bar{h}(l, l_{X'}) \left( \mathbb{P}_{\pi_\xi}(Y = y_l, \bar{Y}^r = 1 | \hat{Y}(X')) - \mathbb{P}_{\pi_\xi}(Y = y_l | \hat{Y}(X')) \mathbb{P}_{\pi_\xi}(\bar{Y}^r = 1 | \hat{Y}(X')) \right) \right|.
\end{aligned}
$$

Let

$$
\begin{aligned}
f_1(l) &= \mathbb{P}_{\pi_\xi}(Y = y_l, \bar{Y}^r = 1 | \hat{Y}(X')) \\
f_2(l) &= \mathbb{P}_{\pi_\xi}(Y = y_l | \hat{Y}(X')) \mathbb{P}_{\pi_\xi}(\bar{Y}^r = 1 | \hat{Y}(X')),
\end{aligned}
$$

and

$$
\begin{aligned}
\mathcal{B}^1 &= \{ l : f_1(l) \geq f_2(l), l \neq l_{X'} \}, \\
\mathcal{B}^2 &= \{ l : f_1(l) < f_2(l), l \neq l_{X'} \}.
\end{aligned}
$$

Therefore,

$$\left| \sum_{l \neq l_{X'}} \bar{h}(l, l_{X'}) \left( \mathbb{P}_{\pi_\xi}(Y = y_l, \bar{Y}^r = 1 | \hat{Y}(X')) - \mathbb{P}_{\pi_\xi}(Y = y_l | \hat{Y}(X')) \mathbb{P}_{\pi_\xi}(\bar{Y}^r = 1 | \hat{Y}(X')) \right) \right|$$

$$= \left| \sum_{l \neq l_{X'}} \bar{h}(l, l_{X'}) \left( f_1(l) - f_2(l) \right) \right|$$

$$= \left| \sum_{l \in \mathcal{B}^1} \bar{h}(l, l_{X'}) \left( f_1(l) - f_2(l) \right) + \sum_{l \in \mathcal{B}^2} \bar{h}(l, l_{X'}) \left( f_1(l) - f_2(l) \right) \right|$$

$$\leq \max_{t=1,2} \left| \sum_{l \in \mathcal{B}^\mathsf{T}} \bar{h}(l, l_{X'}) \left( f_1(l) - f_2(l) \right) \right|$$

$$= \max \left\{ \sum_{l \in \mathcal{B}^1} \bar{h}(l, l_{X'}) \left( f_1(l) - f_2(l) \right), \sum_{l \in \mathcal{B}^2} \bar{h}(l, l_{X'}) \left( f_2(l) - f_1(l) \right) \right\}.$$

On the other hand,

$$\sum_{l \in \mathcal{B}^1} \bar{h}(l, l_{X'}) \left( f_1(l) - f_2(l) \right)$$

$$= \sum_{l \in \mathcal{B}^1} \bar{h}(l, l_{X'}) \left( \mathbb{P}_{\pi_\xi}(Y = y_l, \bar{Y}^r = 1 | \hat{Y}(X')) - \mathbb{P}_{\pi_\xi}(Y = y_l | \hat{Y}(X')) \mathbb{P}_{\pi_\xi}(\bar{Y}^r = 1 | \hat{Y}(X')) \right)$$

$$\leq 2H \sum_{l \in \mathcal{B}^1} \left( \mathbb{P}_{\pi_\xi}(Y = y_l, \bar{Y}^r = 1 | \hat{Y}(X')) - \mathbb{P}_{\pi_\xi}(Y = y_l | \hat{Y}(X')) \mathbb{P}_{\pi_\xi}(\bar{Y}^r = 1 | \hat{Y}(X')) \right)$$

$$= 2H \left( \mathbb{P}_{\pi_\xi}(Y = y_l, \exists l \in \mathcal{B}^1, \bar{Y}^r = 1 | \hat{Y}(X')) - \mathbb{P}_{\pi_\xi}(Y = y_l, \exists l \in \mathcal{B}^1 | \hat{Y}(X')) \mathbb{P}_{\pi_\xi}(\bar{Y}^r = 1 | \hat{Y}(X')) \right)$$

$$= 2H \left( \mathbb{P}_{\pi_\xi}(Y = y_l, \exists l \in \mathcal{B}^1, \bar{Y}^r = 1 | \hat{Y}(X')) \mathbb{P}_{\pi_\xi}(\bar{Y}^r = 0 | \hat{Y}(X')) \right.$$

$$\left. - \mathbb{P}_{\pi_\xi}(Y = y_l, \exists l \in \mathcal{B}^1, \bar{Y}^r = 0 | \hat{Y}(X')) \mathbb{P}_{\pi_\xi}(\bar{Y}^r = 1 | \hat{Y}(X')) \right)$$

$$\leq 2H \left( \mathbb{P}_{\pi_\xi}(\bar{Y}^r = 1 | \hat{Y}(X')) \mathbb{P}_{\pi_\xi}(\bar{Y}^r = 0 | \hat{Y}(X')) \right)$$

$$= 2H \mathbf{Var} \left[ \bar{Y}^r | \hat{Y}(X') \right].$$

Similarly, we have

$$\sum_{l \in \mathcal{B}^1} \bar{h}(l, l_{X'}) \left( f_2(l) - f_1(l) \right) - \sum_{l \in \mathcal{B}^2} \bar{h}(l, l_{X'}) \left( \mathbb{P}_{\pi_\xi}(Y = y_l, \bar{Y}^r = 1 | \hat{Y}(X')) + \mathbb{P}_{\pi_\xi}(Y = y_l | \hat{Y}(X')) \mathbb{P}_{\pi_\xi}(\bar{Y}^r = 1 | \hat{Y}(X')) \right)$$

$$\leq 2H \mathbf{Var} \left[ \bar{Y}^r | \hat{Y}(X') \right].$$

Conclusively, we have

$$\left| \frac{\partial g_{X'}(\xi)}{\partial \theta_{\hat{y}_r, j}^{\text{Acc}}} \right| \leq 2H \mathbf{Var} \left[ \bar{Y}^r | \hat{Y}(X') \right].$$

If $\hat{Y}^j = \hat{y}_r$, similar to (1), we have

$$\left| \frac{\partial g_{X'}(\xi)}{\partial \theta_{\hat{y}_r, j}^{\text{Acc}}} \right| = 0.$$

(3). For any $\hat{y}_i, \hat{y}_j \in \hat{\mathcal{Y}}$, by the definition of $\phi^t_{\hat{y}_i,\hat{y}_j}$, there exists $(a,b) \in \{0,1\}^2$, such that $\phi^t_{\hat{y}_i,\hat{y}_j}(a,b) \neq 0$. Similar to (2), let

$$f_3(l) = \mathbb{P}_{\pi_\xi}(Y = y_l, \bar{Y}^i = a, \bar{Y}^j = b|\hat{Y}(X'))$$
$$f_4(l) = \mathbb{P}_{\pi_\xi}(Y = y_l|\hat{Y}(X'))\mathbb{P}_{\pi_\xi}(\bar{Y}^i = a, \bar{Y}^j = b|\hat{Y}(X'))$$

and

$$\mathcal{B}^3 = \{l : f_3(l) \geq f_4(l), l \neq l_{X'}\},$$
$$\mathcal{B}^4 = \{l : f_3(l) < f_4(l), l \neq l_{X'}\}$$

we have

$$\left|\frac{\partial g_{X'}(\xi)}{\partial \theta^t_{\hat{y}_i,\hat{y}_j}}\right| = \max\left\{\sum_{l\in\mathcal{B}^3} \bar{h}(l, l_{X'})(f_3(l) - f_4(l)), \sum_{l\in\mathcal{B}^4} \bar{h}(l, l_{X'})(f_4(l) - f_3(l))\right\}$$
$$\leq 2H\mathbf{Var}\left[\phi^t_{\hat{y}_i,\hat{y}_j}(\bar{Y}^i, \bar{Y}^j)|\hat{Y}(X')\right]$$
$$\overset{(*)}{\leq} 2H\left(\mathbf{Var}\left[\bar{Y}^i|\hat{Y}(X')\right] + \mathbf{Var}\left[\bar{Y}^j|\hat{Y}(X')\right]\right),$$

where inequality $(*)$ comes from Lemma D.1.

(4). For any $y_i \in \mathcal{Y}, \hat{y}_r \in \hat{\mathcal{Y}}$, by the definition of $\phi^t_{y_i,\hat{y}_r}$, there exists $a \in \{0,1\}, y_j \in \mathcal{Y}$, s.t., $\phi^t_{y_i,\hat{y}_r}(y_j, a) \neq 0$. We further divide the proof into two cases: $\phi^t_{y_i,\hat{y}_r}(y_i, a) = 0$, and $\phi^t_{y_i,\hat{y}_r}(y_i, a) \neq 0$.

(4a). If $\phi^t_{y_i,\hat{y}_r}(y_i, a) = 0$, we have $t_{y_i\hat{y}_r} = t^{sg}$ and consequently $a = 1$. Similar to (1-3)., we have

$$\left|\frac{\partial g_{X'}(\xi)}{\partial \theta^t_{y_i,\hat{y}_r}}\right| = \left|\sum_{l\neq l_{X'}} \bar{h}(l, l_{X'})\frac{\partial\mathbb{P}_{\pi_\xi}(y_l|\hat{Y}(X'))}{\partial \theta^t_{y_i,\hat{y}_r}}\right| \overset{(\bullet)}{=} \left|\sum_{l=1}^k \bar{h}(l, l_{X'})\frac{\partial\mathbb{P}_{\pi_\xi}(y_l|\hat{Y}(X'))}{\partial \theta^t_{y_i,\hat{y}_r}}\right|$$
$$= \left|\sum_{l\neq i} \bar{h}(l, l_{X'})\left(\mathbb{P}_{\pi_\xi}(Y = y_l, \bar{Y}^r = 1|\hat{Y}(X')) - \mathbb{P}_{\pi_\xi}(Y = y_l|\hat{Y}(X'))\mathbb{P}_{\pi_\xi}(Y \neq y_i, \bar{Y}^r = 1|\hat{Y}(X'))\right)\right.$$
$$\left. - \bar{h}(i, l_{X'})\mathbb{P}_{\pi_\xi}(Y = y_i|\hat{Y}(X'))\mathbb{P}_{\pi_\xi}(Y \neq y_i, \bar{Y}^r = 1|\hat{Y}(X'))\right|,$$

where Eq. $(\bullet)$ is due to Let

$$f_5(l) = \mathbb{P}_{\pi_\xi}(Y = y_l, \bar{Y}^r = 1|\hat{Y}(X'))$$
$$f_6(l) = \mathbb{P}_{\pi_\xi}(Y = y_l|\hat{Y}(X'))\mathbb{P}_{\pi_\xi}(Y \neq y_i, \bar{Y}^r = 1|\hat{Y}(X'))$$

and

$$\mathcal{B}^5 = \{l : f_5(l) \geq f_6(l), \quad l \neq i\},$$
$$\mathcal{B}^6 = \{l : f_5(l) < f_6(l), \quad l \neq i\}$$

Then we have

$$\left|\frac{\partial g_{X'}(\xi)}{\partial \theta^t_{y_i,\hat{y}_r}}\right| \leq \max\left\{\sum_{l\in\mathcal{B}^5} \bar{h}(l, l_{X'})(f_5(l) - f_6(l)), \sum_{l\in\mathcal{B}^6} \bar{h}(l, l_{X'})(f_6(l) - f_5(l)) + \bar{h}(i, l_{X'})f_6(i)\right\}.$$

On one hand,

$$\sum_{l \in \mathcal{B}^5} \bar{h}(l, l_{X'}) \left(f_5(l) - f_6(l)\right)$$

$$= \sum_{l \in \mathcal{B}^5} \bar{h}(l, l_{X'}) \left(\mathbb{P}_{\pi_\xi}(Y = y_l, \bar{Y}^r = 1|\hat{Y}(X')) - \mathbb{P}_{\pi_\xi}(Y = y_l|\hat{Y}(X'))\mathbb{P}_{\pi_\xi}(Y \neq y_i, \bar{Y}^r = 1|\hat{Y}(X'))\right)$$

$$\leq \sum_{l \in \mathcal{B}^5} 2H \left(\mathbb{P}_{\pi_\xi}(Y = y_l, \bar{Y}^r = 1|\hat{Y}(X')) - \mathbb{P}_{\pi_\xi}(Y = y_l|\hat{Y}(X'))\mathbb{P}_{\pi_\xi}(Y \neq y_i, \bar{Y}^r = 1|\hat{Y}(X'))\right)$$

$$= 2H \left(\mathbb{P}_{\pi_\xi}(Y = y_l, \exists l \in \mathcal{B}^5, \bar{Y}^r = 1|\hat{Y}(X')) - \mathbb{P}_{\pi_\xi}(Y = y_l, \exists l \in \mathcal{B}^5|\hat{Y}(X'))\mathbb{P}_{\pi_\xi}(Y \neq y_i, \bar{Y}^r = 1|\hat{Y}(X'))\right)$$

$$= 2H \left(\mathbb{P}_{\pi_\xi}(Y = y_l, \exists l \in \mathcal{B}^5, \bar{Y}^r = 1|\hat{Y}(X'))(1 - \mathbb{P}_{\pi_\xi}(Y \neq y_i, \bar{Y}^r = 1|\hat{Y}(X'))) \right.$$
$$\left. - \mathbb{P}_{\pi_\xi}(Y = y_l, \exists l \in \mathcal{B}^5, \bar{Y}^r = 0|\hat{Y}(X'))\mathbb{P}_{\pi_\xi}(Y \neq y_i, \bar{Y}^r = 1|\hat{Y}(X'))\right)$$

$$\leq 2H\mathbb{P}_{\pi_\xi}(Y \neq y_i, \bar{Y}^r = 1|\hat{Y}(X'))(1 - \mathbb{P}_{\pi_\xi}(Y \neq y_i, \bar{Y}^r = 1|\hat{Y}(X')))$$

$$= 2H\mathbf{Var}_{\pi_\xi} \left[\phi^t_{y_i, \hat{y}_l}(Y, \bar{Y}^r)|\hat{Y}(X')\right]$$

$$\leq 2H\mathbf{Var}_{\pi_\xi} \left[\mathbb{1}_{Y = y_i}|\hat{Y}(X')\right] + 2H\mathbf{Var}_{\pi_\xi} \left[\bar{Y}^r|\hat{Y}(X')\right].$$

On the other hand,

$$\sum_{l \in \mathcal{B}^6} \bar{h}(l, l_{X'}) \left(f_6(l) - f_5(l)\right) + \bar{h}(i, l_{X'})f_6(i)$$

$$= - \sum_{l \in \mathcal{B}^6} \bar{h}(l, l_{X'}) \left(\mathbb{P}_{\pi_\xi}(Y = y_l, \bar{Y}^r = 1|\hat{Y}(X')) + \mathbb{P}_{\pi_\xi}(Y = y_l|\hat{Y}(X'))\mathbb{P}_{\pi_\xi}(Y \neq y_i, \bar{Y}^r = 1|\hat{Y}(X'))\right)$$
$$+ \bar{h}(i, l_{X'})\mathbb{P}_{\pi_\xi}(Y = y_i|\hat{Y}(X'))\mathbb{P}_{\pi_\xi}(Y \neq y_i, \bar{Y}^r = 1|\hat{Y}(X'))$$

$$\leq 2H \left(-\mathbb{P}_{\pi_\xi}(Y = y_l, \exists l \in \mathcal{B}^6, \bar{Y}^r = 1|\hat{Y}(X'))(1 - \mathbb{P}_{\pi_\xi}(Y \neq y_i, \bar{Y}^r = 1|\hat{Y}(X'))) \right.$$
$$- \mathbb{P}_{\pi_\xi}(Y = y_l, \exists l \in \mathcal{B}^6, \bar{Y}^r = 0|\hat{Y}(X'))\mathbb{P}_{\pi_\xi}(Y \neq y_i, \bar{Y}^r = 1|\hat{Y}(X'))$$
$$\left. + \mathbb{P}_{\pi_\xi}(Y = y_i|\hat{Y}(X'))\mathbb{P}_{\pi_\xi}(Y \neq y_i, \bar{Y}^r = 1|\hat{Y}(X'))\right)$$

$$\leq 2H \left(\mathbb{P}_{\pi_\xi}(Y = y_l, \exists l \in \mathcal{B}^6, \bar{Y}^r = 0|\hat{Y}(X'))\mathbb{P}_{\pi_\xi}(Y \neq y_i, \bar{Y}^r = 1|\hat{Y}(X')) \right.$$
$$\left. + \mathbb{P}_{\pi_\xi}(Y = y_i|\hat{Y}(X'))\mathbb{P}_{\pi_\xi}(Y \neq y_i, \bar{Y}^r = 1|\hat{Y}(X'))\right)$$

$$\leq 2H \left(\mathbb{P}_{\pi_\xi}(\bar{Y}^r = 0|\hat{Y}(X'))\mathbb{P}_{\pi_\xi}(\bar{Y}^r = 1|\hat{Y}(X')) + \mathbb{P}_{\pi_\xi}(Y = y_i|\hat{Y}(X'))\mathbb{P}_{\pi_\xi}(Y \neq y_i, \bar{Y}^r = 1|\hat{Y}(X'))\right)$$

$$\leq 2H\mathbf{Var}_{\pi_\xi} \left[\mathbb{1}_{Y = y_i}|\hat{Y}(X')\right] + 2H\mathbf{Var}_{\pi_\xi} \left[\bar{Y}^r|\hat{Y}(X')\right].$$

Therefore, in this case, we have

$$\left|\frac{\partial g_{X'}(\xi)}{\partial \theta^t_{y_i, \hat{y}_r}}\right| \leq 2H\mathbf{Var}_{\pi_\xi} \left[\mathbb{1}_{Y = y_i}|\hat{Y}(X')\right] + 2H\mathbf{Var}_{\pi_\xi} \left[\bar{Y}^r|\hat{Y}(X')\right].$$

(4b). If $\phi^t_{y_i, \hat{y}_r}(y_i, a) \neq 0$, similar to $(4a)$., we have

$$\left|\frac{\partial g_{X'}(\xi)}{\partial \theta^t_{y_i, \hat{y}_r}}\right| = \left|- \sum_{l \neq i} \bar{h}(l, l_{X'})\mathbb{P}_{\pi_\xi}(Y = y_l|\hat{Y}(X'))\mathbb{P}_{\pi_\xi}(Y = y_i, \bar{Y}^r = 1|\hat{Y}(X')) \right.$$
$$\left. + \bar{h}(i, l_{X'}) \left(\mathbb{P}_{\pi_\xi}(Y = y_i, \bar{Y}^l = 1|\hat{Y}(X')) - \mathbb{P}_{\pi_\xi}(Y = y_i|\hat{Y}(X'))\mathbb{P}_{\pi_\xi}(Y = y_i, \bar{Y}^r = 1|\hat{Y}(X'))\right)\right|.$$

Since

$$\bar{h}(i, l_{X'}) \left( \mathbb{P}_{\pi_\xi}(Y = y_i, \bar{Y}^r = 1|\hat{Y}(X')) - \mathbb{P}_{\pi_\xi}(Y = y_i|\hat{Y}(X'))\mathbb{P}_{\pi_\xi}(Y = y_i, \bar{Y}^r = 1|\hat{Y}(X')) \right)$$
$$= \bar{h}(i, l_{X'})\mathbb{P}_{\pi_\xi}(Y = y_i, \bar{Y}^r = 1|\hat{Y}(X'))\mathbb{P}_{\pi_\xi}(Y \neq y_i|\hat{Y}(X'))$$
$$\geq 0,$$

we have

$$\left| \frac{\partial g_{X'}(\xi)}{\partial \theta^t_{y_i, \hat{y}_l}} \right| \leq \max \left\{ \bar{h}(i, l_{X'})\mathbb{P}_{\pi_\xi}(Y = y_i, \bar{Y}^r = 1|\hat{Y}(X'))\mathbb{P}_{\pi_\xi}(Y \neq y_i|\hat{Y}(X')) , \right.$$

$$\left. \sum_{l \neq i} \bar{h}(l, l_{X'})\mathbb{P}_{\pi_\xi}(Y = y_l|\hat{Y}(X'))\mathbb{P}_{\pi_\xi}(Y = y_i, \bar{Y}^r = 1|\hat{Y}(X')) \right\}$$

$$\leq 2H\mathbf{Var}\left[ \mathbb{1}_{Y=y_i}|\hat{Y}(X') \right].$$

Combining (4a). and (4b)., we have that

$$\left| \frac{\partial g_{X'}(\xi)}{\partial \theta^t_{y_i, \hat{y}_l}} \right| \leq 2H \left( \mathbf{Var}_{\pi_\xi}\left[ \mathbb{1}_{Y=y_i}|\hat{Y}(X') \right] + \mathbf{Var}_{\pi_\xi}\left[ \bar{Y}^l|\hat{Y}(X') \right] \right).$$

Combining (1-4)., we then have

$$\|\nabla g_{X'}(\xi)\|^2$$
$$\leq 4H^2 \sum_{i=1}^{k} \sum_{j=1}^{n} (|\mathcal{N}(y_i, \mathcal{Y}_j)| - 1) \mathbf{Var}_{\pi_\xi}\left[ \mathbb{1}_{Y=y_i}|\hat{Y}(X') \right]^2 \tag{27}$$

$$+ 4H^2 \sum_{j \in [n], \hat{y}_r \in \mathcal{Y}_j} \mathbf{Var}_{\pi_\xi}\left[ \bar{Y}^r|\hat{Y}(X') \right]^2$$

$$+ 4H^2 \sum_{i,j \in [\hat{k}]} \left( \mathbf{Var}_{\pi_\xi}\left[ \bar{Y}^i|\hat{Y}(X') \right] + \mathbf{Var}_{\pi_\xi}\left[ \bar{Y}^j|\hat{Y}(X') \right] \right)^2$$

$$+ 4H^2 \sum_{i \in [k], j \in [\hat{k}]} \left( \mathbf{Var}_{\pi_\xi}\left[ \mathbb{1}_{Y=y_i}|\hat{Y}(X') \right] + \mathbf{Var}_{\pi_\xi}\left[ \bar{Y}^l|\hat{Y}(X') \right] \right)^2$$

$$\leq 8H^2 \left( \sum_{i=1}^{k} (n_i + \hat{k})\mathbf{Var}_{\pi_\xi}(\mathbb{1}_{Y=y_i}|\hat{Y} = \hat{Y}^*)^2 + \sum_{i=1}^{\hat{k}} (m_i + K - 1)\mathbf{Var}_{\pi_\xi}(\bar{Y}^i|\hat{Y} = \hat{Y}^*)^2 \right). \tag{28}$$

Therefore, by Eqs. (25), (26), and (28), and Assumption Eq. (21), we have

$$|\ell(W) - \ell_{\hat{\Theta}}(W)| = \left| \mathbb{E}_{\pi^*}\left[ \sum_{l \neq l_{X'}} \bar{h}(l, l_{X'}) \left( \mathbb{P}_{\pi_{\Theta^*}}(Y = y_l|\hat{Y}(X')) - \mathbb{P}_{\pi_{\hat{\Theta}}}(Y = y_l|\hat{Y}(X')) \right) \right] \right|$$

$$= \left| \mathbb{E}_{\pi^*}\left[ g_{X'}(\Theta^*) - g_{X'}(\hat{\Theta}) \right] \right|$$

$$\leq \left| \mathbb{E}_{\pi^*}\|\nabla g_{X'}(\xi)\|\|\Theta^* - \hat{\Theta}\| \right|$$

$$\leq \frac{2cH}{\sqrt{M}}\|\Theta^* - \hat{\Theta}\|.$$

Now, we apply the assumption that we are able to solve the empirical problem, producing an estimate $\hat{W}$ that satisfies

$$\mathbb{E}\left[ \ell_{\hat{\Theta}}(\hat{W}) - \ell_{\hat{\Theta}}(W^*_{\hat{\Theta}}) \right] \leq \chi,$$

where $W_{\hat{\Theta}}^*$ is the true solution to

$$W_{\hat{\Theta}}^* = \arg\min_W \ell_{\Theta}(W).$$

Therefore,

$$
\begin{aligned}
\mathbb{E}\left[\ell(\hat{W}) - \ell(W^*)\right] &= \mathbb{E}\left[\ell_{\hat{\Theta}}(\hat{W}) - \ell_{\hat{\Theta}}(W_{\hat{\Theta}}^*) + \ell_{\hat{\Theta}}(W_{\hat{\Theta}}^*) - \ell_{\hat{\Theta}}(\hat{W}) + \ell(\hat{W}) - \ell(W^*)\right] \\
&\overset{(*)}{\leq} \chi + \mathbb{E}\left[\ell_{\hat{\Theta}}(W_{\hat{\Theta}}^*) - \ell_{\hat{\Theta}}(\hat{W}) + \ell(\hat{W}) - \ell(W^*)\right] \\
&\leq \chi + 4cH\frac{1}{\sqrt{M}}\mathbb{E}\|\hat{\Theta} - \Theta^*\| + \mathbb{E}\left[\ell_{\hat{\Theta}}(W_{\hat{\Theta}}^*) - \ell_{\hat{\Theta}}(\hat{W}) + \ell_{\hat{\Theta}}(\hat{W}) - \ell_{\hat{\Theta}}(W^*)\right] \\
&\leq \chi + 4cH\frac{1}{\sqrt{M}}\mathbb{E}\|\hat{\Theta} - \Theta^*\|,
\end{aligned}
$$

where Eq. $(*)$ comes from condition (23).

With Eqs. (20) and (21), we have Eq. (24) by Lemma D.5, i.e.,

$$\left(\mathbb{E}\|\hat{\Theta} - \Theta^*\|\right)^2 \leq \mathbb{E}\|\hat{\Theta} - \Theta^*\|^2 \leq \varepsilon^2 M.$$

We can now bound this using the result of Lemma D.6, which results in

$$\mathbb{E}\left[\ell(\hat{W}) - \ell(W^*)\right] \leq \chi + 4cH\epsilon.$$

The proof is completed.

$\square$

# E EXAMPLES AND ILLUSTRATIONS

## E.1 LABEL GRAPH AND LABEL HIERARCHY

Fig 5 shows the mapping between a label hierarchy and the corresponding label graph. Indeed, given the order of labels, any label structure represented as a (directed acyclic graph) DAG can be converted to exact one consistent label graph based on the four types of label relations.

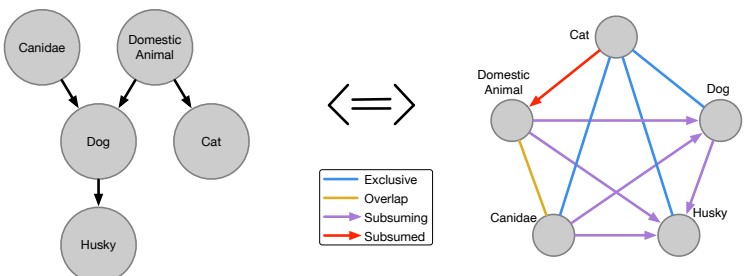

Figure 5: The illustration of mapping between a DAG of labels and a label graph.

## E.2 AN EXAMPLE OF INCONSISTENT LABEL GRAPH

Fig. 6 shows an example of an inconsistent label graph. We can see that the label graph is unrealistic and ambiguous because "*Husky*" subsumes "*Canidae*", but (1) "*Canidae*" subsumes "*Dog*" and (2) "*Dog*" subsuems "*Husky*" combined imply that "*Husky*" should be subsumed by "*Canidae*". Also, from the example, we can see that label graph induced from cyclic label hierarchy must be inconsistent.

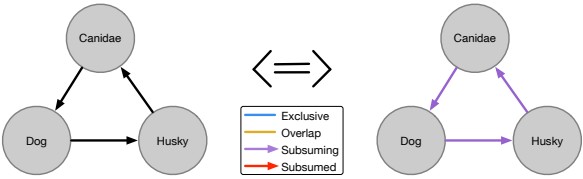

Figure 6: The illustration of inconsistent label graph.

### E.3 ENUMERATION OF INCONSISTENT TRIANGLE LABEL GRAPH

For a triangle label graph $G$, we list all inconsistent label relation structures. The consistency of larger label graph with more labels can be verified by checking the consistency of every triangle inside. One example proof of {Exclusive, Overlap, Subsuming} can be found in Lemma 1.

Table 5: Enumeration of Inconsistent Label Relation Triplets.

| label relation Triplets | | |
|---|---|---|
| $t_{ab}$ | $t_{bc}$ | $t_{ac}$ |
| Overlap | Subsumed | Subsuming |
| Overlap | Subsumed | Exclusive |
| Overlap | Subsuming | Subsumed |
| Overlap | Exclusive | Subsumed |
| Exclusive | Subsumed | Subsuming |
| Exclusive | Overlap | Subsuming |
| Exclusive | Subsuming | Subsuming |
| Exclusive | Subsuming | Subsumed |
| Exclusive | Subsuming | Overlap |
| Subsuming | Exclusive | Subsumed |
| Subsuming | Subsumed | Exclusive |
| Subsuming | Overlap | Subsumed |
| Subsuming | Overlap | Exclusive |
| Subsuming | Subsuming | Exclusive |
| Subsuming | Subsuming | Subsumed |
| Subsuming | Subsuming | Overlap |
| Subsumed | Overlap | Subsuming |
| Subsumed | Subsumed | Exclusive |
| Subsumed | Subsumed | Subsuming |
| Subsumed | Subsumed | Overlap |
| Subsumed | Exclusive | Subsuming |
| Subsumed | Exclusive | Subsumed |
| Subsumed | Exclusive | Overlap |

### E.4 AN EXAMPLE OF INDISTINGUISHABLE LABEL GRAPH

Fig. 7 shows an example label graph with indistinguishable label relation structure. Again, red labels represent desired unseen labels, while gray labels are undesired and seen. We can see that unseen label "*Husky*" and "*Bulldog*" have indistinguishable label relation structures because for all seen labels, their label relations are equal. For example, seen label "*Dog*" subsumes both "*Husky*" and "*Bulldog*". In contrast, for "*Husky*" and "*Bengal Cat*", seen label "*Cat*" subsumes the latter but exclusive to the former, which indicates that "*Husky*" and "*Bengal Cat*" have distinguishable label relation structure. Note that "*Bengal Cat*" and "*Persian Cat*" also have indistinguishable label relation structure, but the former is unseen desired label while the latter is seen and can be predicted by some ILF(s). We are only interested in the distinguishablity of a pair of unseen labels.

In practice, users could "break the symmetry" by adding new ILFs with new labels. For example, if we add an ILF that could predict "*Arctic Animals*", then the new seen label "*Arctic Animals*" will be added into label graph as shown in Fig. 8. We know that "*Arctic Animals*" subsumes "*Husky*" but not "*Bulldog*", so we break the indistinguishable label relation structure of "*Husky*" and "*Bulldog*" successfully.

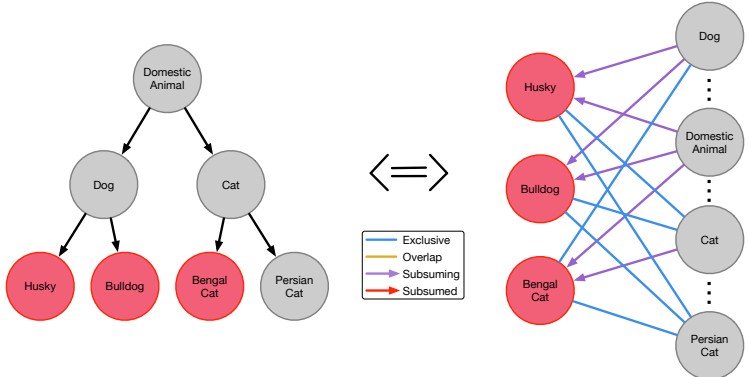

Figure 7: An example of an indistinguishable label relation structure ("*Husky*" and "*Bulldog*").

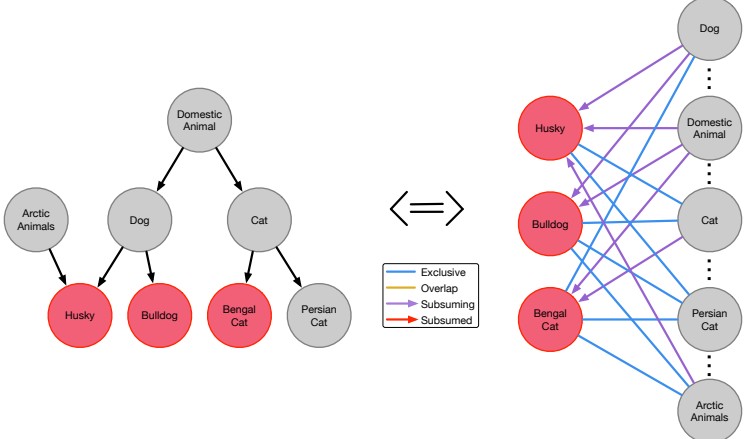

Figure 8: An example of fixing an indistinguishable label relation structure ("*Husky*" and "*Bulldog*") by adding a new label ("*Arctic Animals*").

## F  EXPERIMENTAL DETAILS

### F.1  DATASET

**Large scale Text Classification Dataset**[3]**:** LSHTC-3 (Partalas et al., 2015), a large scale hierarchical text classification dataset, which consists of 456,886 documents and 36,504 categories organized in a label hierarchy. We filter out the documents with multiple labels, and preserve categories with more than 500 documents. We use a pre-trained sentence transformer (Reimers & Gurevych, 2019) to obtain document embeddings for classification. We follow Zhang et al. (2021) to generate 5 keyword-based labeling functions for each seen label as ILFs.

**Large scale Image Classification Dataset**[4]**:** ILSVRC2012 (Russakovsky et al., 2015), a large scale image classification dataset, which consists of 1.2M training images from 1000 object classes based

---

[3]http://lshtc.iit.demokritos.gr/
[4]http://image-net.org/challenges/LSVRC/2012/index#data

on ImageNet. Following Deng et al. (2014) we use WordNet as the label hierarchy, and because all the images are assigned to leave labels in WordNet, for each non-leave label, we aggregate images belonging to its descendants as its data points (Deng et al., 2014). For weak supervision sources creation, we follow Mazzetto et al. (2021b;a) to train 10 image classifiers as ILFs. We randomly sampling 2 or 3 exclusive seen labels from the label graph as well as 500 images for each label to train a ResNet-32 classifier.

## F.2   DESCRIPTION OF APPLYING DAP

To apply DAP, we use both label relations and ILFs to construct attributes for both unseen classes and unlabeled data points. Then, we train the attribute classifiers, which in turn are used to predict unseen labels on the test set as in Lampert et al. (2013). To construct attributes for unseen labels and data points, we leverage the outputs of ILFs and label relations.

First, based on the label relations and basic logistic rules, we enumerate all the possible assignments of seen labels given a data point. For example, if label $A$ is subsumed by label $B$, then for a data point, when it belongs to label $A$, it must also belong to $B$; And if label $A$ and $B$ are exclusive, then one data cannot belong to both at the same time. Let $s \in S$ denote one possible label assignment and $S$ is the set of all possible $s$. Then we define the attribute as a vector of $|S|$ dimension where each dimension corresponds to one $s$.

Second, we define the attribute of unseen labels. For an unseen label $A$ and a label assignment $s$, if $A$ is not exclusive to any label in $s$ then we set the corresponding attribute $a_s = 1$ for label $A$, other wise 0. The intuition is that, if $A$ is not exclusive to labels in $s$, it's likely that when a data belongs to assignment $s$, it also belongs to label $A$. For each data point, we use the labels assigned by ILFs to build their attributes. If a data belongs to assignment $s$ then its corresponding attribute $a_s = 1$, otherwise 0.

Then, we can train attribute classifier $p(a|x)$ for each attribute based on data point attributes. During inference, we use unseen label attribute as well as attribute classifier as in Lampert et al. (2013):

$$f(x) = \arg\max_c \prod_{m=1}^{|S|} \frac{p(a_m^c|x)}{p(a_m^c|x)} \tag{29}$$

## F.3   HYPER-PARAMETERS

For the training of PGMs, we set the learning rate to be $\frac{1}{n}$ where $n$ is the number of training data. For training logistic regression model, we use the default parameters in scikit-learn library. For training ResNet model, we set batch size as 256 and use Adam optimizer with learning rate being 1e-3 and weight decay being 5e-5.

## F.4   HARDWARE AND IMPLEMENTATION DETAILS

All experiments ran on a machine with an Intel(R) Xeon(R) CPU E5-2678 v3 with a 512G memory and a GeForce GTX 1080Ti-11GB GPU.

All the code was implemented in Python. We use the standard implementation of the logistic regression model from Python scikit-learn library[5] and the ResNet model from torchvision library[6].

Our code will be released upon the acceptance.

## F.5   DATASET DETAILS OF REAL-WORLD APPLICATIONS

We list the tags we used in the real-world application (Sec. 7.3) and examples of label relations we query from the existing product category taxonomy.

---

[5] https://scikit-learn.org/stable/modules/generated/sklearn.linear_model.LogisticRegression.html
[6] https://pytorch.org/docs/stable/torchvision/models.html

Table 6: The tags and examples of label relations of "*Car Accessories*" category.

| new unseen tags: | "*Performance Modifying Parts*", "*Vehicle Tires & Tire Parts*", "*Car Engines & Engine Parts*" |
|---|---|
| existing tags: | "*Car Modification Parts*", "*Car Parts & Accessories*"
"*Car & Truck Tires*", "*Replacement Car Parts*", "*Car & Truck Wheels*" |
| label relation examples: | "*Replacement Car Parts*" **subsumes** "*Car Engines & Engine Parts*"
"*Car & Truck Tires*" **is subsumed by** "*Vehicle Tires & Tire Parts*" |

Table 7: The tags and examples of label relations of "*Furniture Accessories*" category.

| new unseen tags: | "*Clothing & Shoe Storage*", "*Living Room Furniture*", "*Beds & Headboards*" |
|---|---|
| existing tags: | "*Coffee Tables & End Tables*", "*Entertainment & Media Centers*"
"*Bedroom Furniture*", "*Sofas & Chairs*", "*Mattresses*" |
| label relation examples: | "*Bedroom Furniture*" **subsumes** "*Beds & Headboards*"
"*Sofas & Chairs*" **is subsumed by** "*Living Room Furniture*" |

# G  ADDITIONAL EXPERIMENTS

## G.1  PERFORMANCE DROP WHEN THE DISTINGUISHABLE CONDITION IS VIOLATED

To validate the effectiveness of the distinguishable condition, we drive another 100 WIS tasks from LSHTC-3 dataset where each task has at least one pair of unseen labels sharing exactly the same label relation structure. In Table 8, we report the performance drop on the averaged evaluation results over the 100 WIS tasks with comparison to the numbers in Table 2. Although the two sets of WIS tasks are different and therefore are not individually comparable, the averaged performance drop does indicates that the violation of the distinguishable condition results in undesirable synthesized training labels, which implicitly demonstrates the effectiveness of the distinguishable condition.

Table 8: Performance drop on averaged evaluation results over 100 WIS tasks derived from LSHTC-3 when the distinguishable condition is violated.

| Method | | Accuracy | F1-score |
|---|---|---|---|
| Label Model | LR-MV | -11.49 | -13.83 |
| | W-LR-MV | -11.51 | -13.47 |
| | WS-LG | -9.28 | -8.63 |
| | PLRM | -9.66 | -9.63 |
| End Model | LR-MV | -16.14 | -17.08 |
| | W-LR-MV | -15.27 | -15.97 |
| | WS-LG | -13.13 | -13.78 |
| | PLRM | -13.39 | -14.09 |

