# OpenReview forum: "Creating Training Sets via Weak Indirect Supervision"
_ICLR.cc/2022/Conference — ICLR 2022 Poster_

### Official Review · Reviewer_EDJu · 2021-10-27

**Correctness:** 4
**Technical Novelty And Significance:** 3
**Empirical Novelty And Significance:** 3
**Recommendation:** 8
**Confidence:** 4

**Main Review:**

## Strengths

Although similar problem settings (e.g., zero-shot learning, hierarchical classification, or distant supervision) have been studied in the literature, the proposed task seems novel and interesting.
In this problem, the given labels are not necessarily the target classes, but they still provide useful information for the classification task.
This work takes a data-centric perspective and provides a way to make use of these supervision sources.

Technically, the proposed method adopts a two-step method that creates pseudo-labels first then trains a classification model, which makes it compatible with many training techniques in the second step.
The use of an exponential family graph model leads to some theoretical guarantees.
The proposed testable condition can be used for evaluating if a specific supervision source can be helpful for a certain classification task.

Experimentally, the proposed method was evaluated on large-scale image and text classification datasets equipped with label hierarchies and was tested in two real-world scenarios.
The author provided empirical evidence that the proposed method outperforms four baselines on these datasets.

---

## Weaknesses

This paper can benefit significantly from improved writing and organization.
I may have missed something, but a few symbols are difficult to understand and some concepts seem unnecessary.

For example, is it necessary to formulate this problem with "indirect labeling functions"?
It seems that those functions are only queried once for "unlabeled training set", but doesn't it make it just a multi-labeled training set?
This makes the difference between ZSL and WIS less clear.
Although I understand that the data distribution would be different, e.g., to classify instances from a label set $0$, ZSL uses paired data $(X_1, Y_1)$ from the label set $1$, possibly $X_0$, and label relation between label sets $0$ and $1$; while WIS uses tuples $(X_0, Y_1, Y_2, ...)$ with instances from the label set $0$ and their labels from the label sets $1, 2, \dots$

A writing issue is that the formulation with "one-hot vectors", in my opinion, makes the notation unnecessarily complex and less clear.
For example, currently we have $Y \in \\{0, 1\\}^k$, $y \in \mathcal{Y}$, $Y[a] \in \\{0, 1\\}$, $\mathcal{Y}_{\lambda_j}$, $\hat{\mathcal{Y}}$, $\hat{\mathcal{Y}}(y)$, $\mathbf{y}^a \in \\{0, 1\\}$ (not a vector), $\hat{Y} \in \\{0, 1\\}^\hat{k}$, and $\tilde{Y}$.
So it is easy to forget their meanings and their value ranges.
Also, some symbols are indexed by subscripts, some by superscripts, and some by $[]$.
Accompanied with one-hot vector-valued labeling functions, they make this paper hard to follow.
Alternatively, I think "one-hot vector" can be an implementation detail and it might be more understandable to define the label to be an element in a set and define the relationship with the indicator function/Iverson bracket.

---

## Confirmation

Before I proceed to ask more questions about the method itself, please allow me to confirm my understanding of the notation and the problem setting.

Task:

- $X \in \mathcal{X}$: input feature, e.g., an image of a dog
- $Y \in \\{0, 1\\}^k$: one-hot vector of class, e.g., $[1, 0, 0, 0]$
- $y \in \mathcal{Y}$: class name, e.g., "dog" (is $y$ indexed by a number $a$?)
- $Y_i[a]$: I can somehow understand it but it's a bit confusing. There are too many layers of indirection. Please define "corresponding to $y_a$" more clearly.

Supervision:

- $\lambda_j: \mathcal{X} \to \\{0, 1\\}^{k_{\lambda_j}}: X_i \mapsto \Lambda_{ij}$: labeling function
- $\Lambda_{ij} \in \\{0, 1\\}^{k_{\lambda_j}}$: one-hot vector of indirect label, e.g., $[1, 0]$
- $? \in \mathcal{Y}_{\lambda_j}$: label name, e.g., "caninae"
- $\mathcal{T}$: could you explain the meaning or give some examples of label relations _exclusive_, _overlapping_, _subsuming_, and _subsumed_ in Section 4? (Or move Definition 2 and Figure 4 here?)

Please confirm the items listed above and improve the readability if possible.

---

## Questions and comments

### In general

- Besides more flexible training, could you clarify the advantages of the two-step approach?
- Is the exponential family distribution needed for the theoretical guarantees?
- Could you clarify the difference between distinguishability and identifiability with a concrete example?

### Section 5.1

- Which prior work? Is "accuracy dependency" defined elsewhere?
- $y_d$ does not appear in the first equation? (cf. the notation issue mentioned above)
- A very common misuse of the indicator function: It is defined for a set, not for a proposition (Iverson bracket)

### Section 5.2

- How should we interpret Figure 2?
- "Training an End Model" notation issue: should be $\to$, not $\mapsto$
- Are there alternatives to SGD? Is SGD needed for Theorem 1 to hold?
- Eq. (3): What does $\tilde{Y} \sim \hat{\Theta} | \hat{\Lambda}\_i$ mean?
  Do you mean $\tilde{Y} \sim p\_{\hat{\Theta}}(\tilde{Y} | \hat{\Lambda})$?

### Section 6

- Very minor: inconsistent capitalization, and typo "Husy"
- $\forall \Theta > 0$: Isn't $\Theta$ a vector? If so, the order $>$ is not defined
- If it's "a.e.", then such $\Theta$ and $\tilde{\Theta}$ may still exist but have measure zero. Is this enough?
- What's the difference between $\mathbb{P}$ and $p$?

### Extensions

- The current label relation graph is only deterministic. Is it easy to extend it to be probabilistic?
- Is classification with rejection applicable here?
  For example, if we use a dog/cat-classifier on a desk image, the output may be very random and non-informative.
  Is out-of-distribution detection needed in this problem?

---

## Post-rebuttal

I've read other reviews, the author's responses, and the updated paper. The author addressed most of my concerns and answered my questions well. The presentation of the paper has been greatly improved. Hence I would like to increase my rating.

**Summary Of The Paper:**

This paper studied a weakly supervised classification problem, called **weak indirect supervision**, where the supervision signals are from labels that are different from but still informative of the classes.
The author proposed a new two-step method that first creates probabilistic labels using an **exponential family graphical model** based on (1) a set of **indirect labeling functions** (pretrained classifier, heuristic rules, etc.) that output deterministic labels and (2) a given **label relation graph** (ontology graph, knowledge base, etc.) that captures the relations between the observed labels and the target classes, then uses the generated labels to train a classifier for the target classes.
The author provided a theoretical analysis on the requirements of the labeling functions and derived a generalization error bound.
The proposed method was evaluated on semi-synthetic datasets based on the ImageNet dataset for image classification and the LSHTC dataset for text classification.

**Summary Of The Review:**

This paper studied an interesting problem in weakly supervised learning and proposed a novel method based on the indirect labeling functions and label relation graph.
The proposed method was validated theoretically and empirically.
Overall, this paper provided a data-centric approach for improving machine learning systems, which might be useful to the community.
However, there are still a few concerns mentioned above, and the writing and organization can be further improved.
Hence I recommend acceptance of this paper.
If the author can address my concerns well, I would like to raise my rating.

---

> ### Author Response · Authors · 2021-11-21
> **Response to Reviewer EDJu (1)**
>
> We thank you for your detailed and helpful comments, thorough review, and appreciation of the novelty and practicality of the proposed method, which indeed is motivated by unmet supervision needs in real-world machine learning systems. In particular, we thank the reviewer for their suggestions about notational and organizational clarity, and apologize for any difficulty reading the initial draft of the paper. We have worked to improve the notation, clarity, and organization of the main paper in the latest version based on your valuable suggestions. We have responded to your other comments and questions below.
> >**Q1**: Is it necessary to formulate this problem with "indirect labeling functions"?
>
> **R1**:
> We talk about indirect labeling functions (ILF), following the naming convention in prior weak supervision works as cited, to highlight the facts that, unlike in a simple multi-labeled training set:
> - The resulting labels come from distinct user-provided sources, which then affects how we can model the problem by aiming to estimate the accuracies and other parameters of the ILFs and their relations.
> - The resulting labels can be noisy/inaccurate, and can apply multiple potentially conflicting labels to each data point (as in other classic crowdsourcing or weak supervision settings).
>
> >**Q2**: Could you explain the meaning or give some examples of label relations exclusive, overlapping, subsuming, and subsumed in Section 4? (Or move Definition 2 and Figure 4 here?)
>
> **R2**: We have re-organized the paper based on your comment to provide additional explanations about these concepts more upfront.
> >**Q3**: could you clarify the advantages of the two-step approach?
>
> **R3**: There are several advantages in formulating weak supervision problems, and our proposed approach specifically, as a two-stage approach:
> - The flexibility and modularity of the end discriminative model we are aiming to train with the provided weak supervision
> - The ability to provide theoretical analysis and guarantees tractably about the weak supervision modeling
> - A joint modeling approach usually requires applying LFs during inference time, which limits its application when the number of LFs is large and expensive to run, or when the LFs are proprietary and cannot be shared with the end model.
> - Additionally, recent benchmark studies have suggested that two step approaches generally enjoy better performance [1].
>
> We will add further discussion in the paper about this important design decision.
>
> [1] Zhang, Jieyu, et al. "WRENCH: A Comprehensive Benchmark for Weak Supervision." NeurIPS 2021 dataset \& benchmark track.
> >**Q4**: Is the exponential family distribution needed for the theoretical guarantees?
>
> **R4**: Yes, our analysis depends on the exponential family distribution and the specific design of the dependency functions.
> >**Q5**: Could you clarify the difference between distinguishability and identifiability with a concrete example?
>
> **R5**: Here is a simple example to demonstrate the difference: suppose there is a PLGM with three  desired labels, e.g., \{"cat", "dog", "lion"\}. Identifiability then requires if the marginal distribution of the output $\hat{Y}$ of ILFs are given, the conditional distribution of desired labels are fixed up to label swapping. Specifically, for any $\Theta$, if there exists $\tilde{\Theta}$ s.t.,
> $\mathbb{P}\_{\Theta}(\hat{Y})=\mathbb{P}\_{\tilde{\Theta}}(\hat{Y})$
> for all $\hat{Y}$, identifiability requires there exists a permutation $(a,b,c)$ of the desired labels, i.e., $\{a,b,c\}=$\{"cat", "dog", "lion"\}, such that,  $\mathbb{P}\_{\tilde{\Theta}}(Y=a|\bar{Y},\hat{Y})=\mathbb{P}\_{\Theta}(Y="cat"|\bar{Y},\hat{Y})$, $\mathbb{P}\_{\tilde{\Theta}}(Y=b|\bar{Y},\hat{Y})=\mathbb{P}\_{\Theta}(Y="dog"|\bar{Y},\hat{Y})$, and
> $\mathbb{P}\_{\tilde{\Theta}}(Y=c|\bar{Y},\hat{Y})=\mathbb{P}\_{\Theta}(Y="lion"|\bar{Y},\hat{Y})$. On the other hand, with $\mathbb{P}\_{\Theta}(\hat{Y})=\mathbb{P}\_{\tilde{\Theta}}(\hat{Y})$ for all $\hat{Y}$, distinguishability requires for almost everywhere $\Theta$, the conditional distribution of desired labels are different after label swapping (for example, it is impossible that $\mathbb{P}\_{\tilde{\Theta}}(Y="dog"|\bar{Y},\hat{Y})=\mathbb{P}\_{\Theta}(Y="cat"|\bar{Y},\hat{Y})$, $\mathbb{P}\_{\tilde{\Theta}}(Y="cat"|\bar{Y},\hat{Y})=\mathbb{P}\_{\Theta}(Y="dog"|\bar{Y},\hat{Y})$, and
> $\mathbb{P}\_{\tilde{\Theta}}(Y="lion"|\bar{Y},\hat{Y})=\mathbb{P}\_{\Theta}(Y="lion"|\bar{Y},\hat{Y})$).
> >**Q6**: Which prior work? Is "accuracy dependency" defined elsewhere?
>
> **R6**: The reference of the prior work has been added accordingly and we defined the "accuracy dependency" in the Section 5.1 for the first time.
> >**Q7**: yd does not appear in the first equation?
>
> **R7**: We improved the notation and fixed this issue.

---

> ### Author Response · Authors · 2021-11-21
> **Response to Reviewer EDJu (2)**
>
> >**Q8**: A very common misuse of the indicator function: It is defined for a set, not for a proposition (Iverson bracket)
>
> **R8**: We fixed this misuse throughout the paper.
>
>
> >**Q9**: How should we interpret Figure 2?
>
> **R9**: We have moved the Figure 2 to Appendix B with a detailed enumeration of our PLRM model. We hope this could help understand our model better.
>
>
> >**Q10**:"Training an End Model" notation issue
>
> **R10**: We also fixed this notation misuse.
>
>
> >**Q11**: Are there alternatives to SGD? Is SGD needed for Theorem 1 to hold?
>
> **R11**: (a). Good question and thanks for asking. Here the stochastic gradient descent is a little different from the one used in deep learning as the data we use in each iteration is sampled independently from the data distribution, instead of sampled from a pre-sampled dataset. This ensures the output parameter can get arbitrarily close to the ground truth (in expectation) with sufficiently small learning rate. For other optimizers, GD can only ensure the output parameter is close to the optimal parameter of the loss, which may not be the ground truth.
>
> (b). The proof of Theorem 1 relies on the specific update rule of SGD. It is not clear whether the result still holds if SGD is replaced with other stochastic optimizers such as SGD with momentum and Adam, as their theoretical analysis is much more involved.
>
>
> >**Q12**: Eq. (3): What does $Y\sim \Theta|\Lambda$ mean? Do you mean $Y\sim p_{\Theta}(Y|\Lambda)$ ?
>
>
> **R12**: Yes, we improved this part to make it clearer.
>
>
> >**Q13**: : Isn't $\Theta$ a vector? If so, the order > is not define
>
> **R13**: Here we mean elementwisely larger than zero, we fixed this in new draft.
>
>
> >**Q14**: If it's "a.e.", then such  and
>  may still exist but have measure zero. Is this enough?
>
> **R14**: There might be a misunderstanding and here the "a.e." means every $\Theta>0$ (element wisely), with which there exists $\tilde{\Theta}>0$ s.t. Eq. (4-6) hold,  lies in a measure zero set, which can hardly be attained in practice.
>
>
> >**Q15**: What's the difference between $\mathbb{P}$ and $p$?
>
> **R15**: They both means probability. We apologize for the notation inconsistency, we have made sure the notations are consistent in the new draft.
>
> >**Q16**: The current label relation graph is only deterministic. Is it easy to extend it to be probabilistic?
>
> **R16**: Yes. If we know the strength $w$ of label relation of any pair of labels, we could refine the current label relation dependency with the strength. Specifically, we use the $w\times\phi$ as the new dependency function. If we know the conditional probability of each pair of labels $p(y_1|y_2)$, then we only need to estimate the accuracy of ILFs while using $p(y_1|y_2)$ to infer desired labels.
>
> >**Q17**: Is classification with rejection applicable here? For example, if we use a dog/cat-classifier on a desk image, the output may be very random and non-informative. Is out-of-distribution detection needed in this problem?
>
> **R17**: That's a very instructive suggestion and leads to interesting future work. Yes, the  out-of-distribution (OOD) issue do exist in applying ILF and it's quite natural to incorporate techniques from OOD, especially classification with rejection. We really appreciate that you point this out and will consider it as an interesting future work to further improve our system. Thanks again!

---

### Official Review · Reviewer_ZVDL · 2021-11-02

**Correctness:** 4
**Technical Novelty And Significance:** 3
**Empirical Novelty And Significance:** 3
**Recommendation:** 8
**Confidence:** 2

**Main Review:**

Strength:
* The paper proposes an interesting problem setting, weak indirect supervision, which is a practical setting and helpful for reducing annotation costs in certain applications.
* Comprehensive theoretical analysis on error bound and distinguishability. However I'm not an expert in PGMs, so I didn't check the proof in the appendix thoroughly.
* Comprehensive experiments with both synthetic datasets and real-world applications. The proposed PLRM method outperforms baselines by a large margin.

Weakness:
I don't see major weaknesses; here are some suggestions and questions.
* I feel the following two paper may be relevant to the WIS problem setting, please consider adding them to the related work section and provide further discussion.
  * Few-shot Relation Extraction via Bayesian Meta-learning on Task Graphs. ICML 2020. (It studies few-shot/zero-shot transfer to new relations, by leveraging relation graphs, similar to the label graph in this paper.)
  * Co-Tuning for Transfer Learning. NeurIPS 2020. (It studies transferring a model trained on source categories to target categories. Problem setting is quite different as Co-Tuning assumes target training data, however I feel it's still quite relevant.)
* Seems the datasets used all have _hierarchical_ label relations. Does that mean the labels are forming a tree structure? I wonder how often "overlap" relation exists in the sampled relation graphs.
* The sampled label graphs have 8 classes. I wonder how the proposed method scales with more classes.

Suggestions for paper presentation:
* Please consider providing some more examples of the sampled label graphs. I see one in the Appendix E.5, but more examples (perhaps with different structure of graphs) will help me get a sense how hard this task is for humans.
* The notations are a little complicated; it would be great if the authors could mention that there is a glossary in the Appendix A in the beginning.


**Summary Of The Paper:**

The authors summarized their contributions quite well in the introduction section -- In this paper, the authors (1) propose a new problem setting called Weak Indirect Supervision (WIS), which extends from the problem setting of Weak Supervision; (2) develop Probabilistic Label Relation Model (PLRM) for WIS; (3) introduce the concept of distinguishability in WIS; (4) conduct empirical evaluation of PLRM on image/text classification and an advertising application.

**Summary Of The Review:**

Strength: interesting new problem setting; comprehensive theoretical analysis; the proposed method has good performance on three datasets.

Weakness: some potential missing references; paper presentation may be further improved; some more discussions on the questions I raised may be helpful.

---

> ### Author Response · Authors · 2021-11-21
> **Response to Reviewer ZVDL**
>
> We thank you for your valuable feedback and suggestions of potential references!
> We found that the papers you recommended are precise and relevant in terms of label relation. In our latest version, we have added the discussion about the two works in the introduction. We have also worked to improve the clarity of our notations throughout the paper, and now explicitly mention that there is a glossary in the Appendix A in the beginning. We have responded to your other questions and comments below.
>
>
> >**Q1**: Seems the datasets used all have hierarchical label relations. Does that mean the labels are forming a tree structure?
>
> **R1**: The datasets studied in the experiments section in fact all have label relation structures that form directed acyclic graphs (DAGs), except for the real-world commercial advertising system, which is a simple hierarchy.
> In terms of the sampled labels: if there is "overlapping" relation, then the label graph is also a DAG, otherwise it's a tree.
> We have improved our descriptions of the datasets in the experiment section to further clarify this.
>
> >**Q2**: Scaling of the proposed method with more classes.
>
> **R2**: A short answer is that the running time of synthesizing training labels will increase with more classes since more classes lead to more latent variables in a PGM, but the training and inference of the end model given the synthesized labels remains the same as in ordinary training of a classifier with more classes. Specifically, both inference and training of the PGM involve Gibbs sampling, with more classes (and therefore more latent variables), the sampling chain will become longer. However, as our goal is to render an end model, we only have to use the learned PGM to infer the training labels once and use the inferred labels to train the end model. Therefore, for downstream tasks, more classes won't introduce computational burden except for a longer process of synthesizing training labels, which is a one-time cost.
>
>
> >**Q3**: Please consider providing some more examples of the sampled label graphs.
>
> **R3**: Yes, we visualized all the 100 sampled label graphs of the LSHTC dataset. The figures can be found in the attached link. In each figure, the red nodes are desired labels while the blue ones are seen labels; we omitted the exclusive relation and the edge label 1, 2 and 3 correspond to the overlapping, subsuming and subsumed label relation respectively.
> We will include more of these examples in the appendix per the reviewer's helpful suggestion!
> https://drive.google.com/drive/folders/1OYrFjym_NYHv7VYYrOiUz_wR5EqYDIB1?usp=sharing

---

> > ### Comment · Reviewer_ZVDL · 2021-11-30
> > **Reply**
> >
> > Thanks for the clarification and additional figures.
> >
> > As mentioned earlier I don't see major weaknesses in this paper. The authors addressed my concerns in the response. In addition they have updated and refined their paper in the discussion period. I will raise my score accordingly.

---

> > > ### Author Response · Authors · 2021-11-30
> > > **Re: Re: Response to Reviewer ZVDL**
> > >
> > > Thank you! We're happy to see the response help address your concerns!

---

### Official Review · Reviewer_Zxiq · 2021-11-02

**Correctness:** 3
**Technical Novelty And Significance:** 2
**Empirical Novelty And Significance:** 2
**Recommendation:** 6
**Confidence:** 4

**Main Review:**

Strength
* Framing the “Weak Indirect Supervision” problem is quite interesting since it is not too uncommon that we have noisy hierarchical labels.
* Overall the paper is clean and make a reasonable contribution by proposing the probabilistic label relation model along with theoretical analysis.

Weakness
* Label relation should be one of pre-set relationships. In other words, the labels should be “comparable” in some sense and the proposed method (and also probably for the several similar label graph based methods) cannot model properly when there are a few labels that are not in the similar semantic space. For example, {red, green} and {dog, bird} are not comparable and some relations cannot be expressed by the label graph. Hence, it is natural to apply the current technique in hierarchical labels, but not for heterogeneous (or multi-modal) labels. I wanted to ask the authors to clarify and discuss on this topic in the main text.
* Datasets used in the paper is all for the hierarchical label interactions. It feels it's somewhat limited than what the authors claims.
* If I understand correctly, PLRM models the relation from ILF to an label with accuracy levels, which are independent from the actual value of ILF outputs. Hence, the noisy level or the certainty from ILFs for each individual data points are ignored, which seems to be a bit limiting.



**Summary Of The Paper:**

This paper introduces a probabilistic framework that is designed to aggregate multiple weak supervision to a single strong proxy guidance to train a model with insufficient supervision from data. Authors claim the proposed framework differs from the previous attempts since it can also model a non-overlapping set of tasks.

**Summary Of The Review:**

I feel the paper is really at borderline (slightly leaning negative) considering 1) the contribution is a bit limited because the framework is only applicable for hierarchical label space 2) modeling ILF to label space with a graphical model is not too novel 3), but the paper still made reasonable contribution in this limited scope.

-------------

Post-rebuttal comment:

I thank authors' response and the clarification on the "comparable" assumption in their paper. I also agree on other reviewer's points for the contributions of the paper and increased my score.

---

> ### Author Response · Authors · 2021-11-21
> **Response to Reviewer Zxiq**
>
> We thank you for your detailed and helpful comments! We answered your questions as below.
>
> >**Q1-2**: Limitations of the label relations considered.
>
> **R1-2**: The types of label relations considered in our work extend beyond label hierarchies, and in fact encompass general directed acyclic graphs (DAGs) with edges of the four label relation types we describe, which correspond to set relations (see Fig. 4).
> The label relation structures of the datasets studied in our experiments (except for the real-world commercial advertising system) are also DAGs rather than simple trees.
>
> However, we certainly agree that these types of label relations absolutely require the labels to be "comparable", i.e. from the same semantic space, in the way you dsescribes, and have worked to clarify this further in Section 3 to avoid any confusion.  We have also edited usages of the term "label hierarchy" in the text.
>
>
> >**Q3**: The noisy level or the certainty from ILFs for each individual data points are ignored, which seems to be a bit limiting.
>
> **R3**: Following the standard weak supervision paradigm, as in prior works like those we cite in our related work section, we assume that the output of the ILFs are categorical.
> For example, ILFs are often from user-authored heuristics, knowledgebases, or black-box models with categorical outputs only.
> However, we appreciate the reviewer's suggestion to consider ILFs with probabilistic outputs as well, which is a suggestion we will take into consideration for future work.

---

> > ### Comment · Reviewer_Zxiq · 2021-11-26
> > **Re: Response to Reviewer Zxiq**
> >
> > I thank authors clarifying the comparability assumption (or "same semantic space") in their main text.
> >
> > I also read other reviewer's comments and agree on their contribution outweighs some of their limitation. Hence, I am increasing the score.

---

> > > ### Author Response · Authors · 2021-11-26
> > > **Re: Re: Response to Reviewer Zxiq**
> > >
> > > Thank you for your reply! And thank you again for your valuable suggestions!

---

### Official Review · Reviewer_Vocb · 2021-11-03

**Correctness:** 4
**Technical Novelty And Significance:** 3
**Empirical Novelty And Significance:** 3
**Recommendation:** 8
**Confidence:** 4

**Main Review:**

Strengths:
This paper studies an important and novel research problem in the weak supervision(WS) setting. In WS, one wants to use as many sources as possible and in many settings indirect LFs can be easily available. However existing WS frameworks cannot accommodate such indirect LFs. Proposed method is technically sound and backed by analysis of generalization error of the end model and distinguishability of unseen labels. Moreover, the experiments on real-world data are promising as well. Overall, the paper is well written and easy to follow. Examples provided help in understanding the setup better.

Weaknesses:
I can't see any major issues with this work. The generalization error analysis is similar to prior work (Ratner 2016) but I think it is a good sanity check. There are some grammar and spelling mistakes which are listed below. If some details/intuition for Theorem 2 can be provided in the main paper, that would be helpful.

Grammar and spelling errors:
in abstract "....an generalization bound."
in section 6 first para "Husy" is used several times, I think the authors meant "Husky" ??
in section 5.1 "... Specifically, for a ILF λj and one ..."


**Summary Of The Paper:**

The paper addresses a novel research problem of using "indirect" label sources in the weak supervision framework to create labelled datasets. The indirect label sources are similar to the labeling functions in prior works in weak supervision ( data programming) with one caveat that these sources produce labels from different space than that of the original ( target) label spaces. It can be useful, when there are good indirect LFs and there is some relationships between the labels in the LF's label space and target label space. This paper, gives a probabilistic label model (PRML) which utilizes these indirect LFs and label relationships to produce desired labels for the given unlabeled data. The methodology is backed by theoretical analysis and real-world experiments. In analysis, a generalization error bound (for the end model learned using estimated labels) is provided which turns out to be similar to the work in Data programming (Ratner et al. 2016). One needs to be careful with the issue of indistinguishability between labels, in such setup. This issue has been studied in detail and they provide definition, conditions for distinguishability. Experiments on real-world data shows that the proposed method works well in comparison to several competing baselines.

**Summary Of The Review:**

I think, overall its a good paper which solves an important problem that in turn broadens the applicability of weak supervision. The work is novel and technically sound. In my opinion, it can be a clear accept.

---

> ### Author Response · Authors · 2021-11-21
> **Response to Reviewer Vocb**
>
> Thank you for the positive comments- we are very happy to hear that you found the paper interesting! Thank you for catching some grammar/spelling issues- we have fixed these in our latest version.
>
>
> >**Q1**: If some details/intuition for Theorem 2 can be provided in the main paper, that would be helpful.
>
> **R1**: In the latest draft, we have worked to improve our overall notation which we hope will make Theorem 2 and related definitions easier to intuitively understand. Additionally, in Appendix E we have provided several examples and visualizations of concepts related to Theorem 2 for better understanding.

---

### Author Response · Authors · 2021-11-21
**Thank all the reviewers!**

We thank all the reviewers for their helpful feedback on the submission!
Based on the valuable comments and suggestions, we have added some new content to the latest draft (highlighted in blue) and worked to improve the main body, appendix (all the proofs), and overall notation for readability.

We’ve provided replies to individual reviewer comments. Please let us know if you have additional questions. Thanks again!

---

### Decision · Program_Chairs · 2022-01-20

**Decision:**

Accept (Poster)

**Comment:**

Scores ultimately point to accept. The one negative review is borderline and doesn't raise any red flags. Weaknesses in other reviews mainly point to minor improvements in the paper and are largely supportive. Rebuttal points are uncontroversial and seem to clarify several issues.